# Essentiality of c-di-AMP in *Bacillus subtilis*: Bypassing mutations converge in potassium and glutamate homeostasis

**Larissa Krüger**[1], **Christina Herzberg**[1], **Hermann Rath**[2], **Tiago Pedreira**[1], **Till Ischebeck**[3], **Anja Poehlein**[4], **Jan Gundlach**[1], **Rolf Daniel**[4], **Uwe Völker**[2], **Ulrike Mäder**[2], **Jörg Stülke**[1]*

**1** Department of General Microbiology, GZMB, Georg-August-University Göttingen, Göttingen, Germany, **2** Interfaculty Institute for Genetics and Functional Genomics, University Medicine Greifswald, Greifswald, Germany, **3** Department of Plant Biochemistry, GZMB, Georg-August-University Göttingen, Göttingen, Germany, **4** Department of Genomic and Applied Microbiology, GZMB, Georg-August-University Göttingen, Göttingen, Germany

* jstuelk@gwdg.de

**Data Availability Statement:** The transcriptomic data reported in this study have been submitted to the NCBI Gene Expression Omnibus (GEO; http://www.ncbi.nlm.nih.gov/geo/) under accession

## Abstract

In order to adjust to changing environmental conditions, bacteria use nucleotide second messengers to transduce external signals and translate them into a specific cellular response. Cyclic di-adenosine monophosphate (c-di-AMP) is the only known essential nucleotide second messenger. In addition to the well-established role of this second messenger in the control of potassium homeostasis, we observed that glutamate is as toxic as potassium for a c-di-AMP-free strain of the Gram-positive model bacterium *Bacillus subtilis*. In this work, we isolated suppressor mutants that allow growth of a c-di-AMP-free strain under these toxic conditions. Characterization of glutamate resistant suppressors revealed that they contain pairs of mutations, in most cases affecting glutamate and potassium homeostasis. Among these mutations, several independent mutations affected a novel glutamate transporter, AimA (Amino acid importer A, formerly YbeC). This protein is the major transporter for glutamate and serine in *B. subtilis*. Unexpectedly, some of the isolated suppressor mutants could suppress glutamate toxicity by a combination of mutations that affect phospholipid biosynthesis and a specific gain-of-function mutation of a mechanosensitive channel of small conductance (YfkC) resulting in the acquisition of a device for glutamate export. Cultivation of the c-di-AMP-free strain on complex medium was an even greater challenge because the amounts of potassium, glutamate, and other osmolytes are substantially higher than in minimal medium. Suppressor mutants viable on complex medium could only be isolated under anaerobic conditions if one of the two c-di-AMP receptor proteins, DarA or DarB, was absent. Also on complex medium, potassium and osmolyte toxicity are the major bottlenecks for the growth of *B. subtilis* in the absence of c-di-AMP. Our results indicate that the essentiality of c-di-AMP in *B. subtilis* is caused by the global impact of the second messenger nucleotide on different aspects of cellular physiology.

number GSE156738. All numerical data that underlies graphs or summary statistics are provided in S4 Table.

**Funding:** This work was supported by a grant from the Deutsche Forschungsgemeinschaft (STU214/16-2) via SPP 1879. The funders had no role in study design, data collection and analysis, decision to publish, or preparation of the manuscript.

**Competing interests:** The authors have declared that no competing interests exist.

## Author summary

Bacteria are exposed to constantly changing environmental conditions. In order to respond to these changes, they use nucleotide second messengers to transduce external signals and translate them into a specific cellular response. Among the repertoire of bacterial second messenger nucleotides, cyclic di-AMP (c-di-AMP) stands out as it is the only second messenger that is essential for the bacteria that produce it, including the Gram-positive model organism *Bacillus subtilis*. C-di-AMP plays a major role in the control of potassium homeostasis, and we found that glutamate is toxic to a *B. subtilis* strain lacking c-di-AMP to the same extent as potassium. These toxic conditions were the starting point for an extensive suppressor analysis, which led to the identification of a novel glutamate transporter (AimA). If the *B. subtilis* strain lacking c-di-AMP was cultivated on complex medium, the isolation of suppressor mutants was only possible under anaerobic conditions and if either of the two c-di-AMP-binding signal transduction proteins was absent. This suggests that these proteins are a major burden for the cell on complex medium in their c-di-AMP free state. Our result underline the complexity of c-di-AMP signaling and propose new directions for research.

## Introduction

To achieve appropriate responses to changing environmental conditions, all organisms use second messengers which are formed in response to an environmental cue. In bacteria, specific mono- and dinucleotides play a particularly important role as second messengers [1]. The common nucleotide second messengers cyclic AMP, cyclic di-GMP (c-di-GMP), and guanosine tetra/pentaphosphate ((p)ppGpp) are used to prioritize the utilization of carbon sources and to regulate virulence, to control the choice between sessile and motile lifestyles, and to downregulate essential cellular functions upon amino acid starvation, respectively [2–7]. The more recently discovered second messenger cyclic di-AMP is unique in several respects. First, this nucleotide is essential in many bacteria that produce it, suggesting that this molecule plays a role in central cellular activities. Essentiality of c-di-AMP has been observed in many Firmicutes, such as *Bacillus subtilis*, *Listeria monocytogenes*, *Streptococcus agalactiae*, and *Staphylococcus aureus*, but also in the genome-reduced bacterium *Mycoplasma pneumoniae* and in *Chlamydia trachomatis* [8–14]. Second, c-di-AMP is toxic if it accumulates in the cells, and in many bacteria this accumulation results in strongly increased stress sensititvity [9,15–19]. Third, c-di-AMP can bind to proteins and the mRNA molecules encoding them to control both the expression and the activity of certain proteins, as in the case of the *B. subtilis* potassium transporters KtrAB and KimA [20–23]. Last, but not least, c-di-AMP is not intrinsically present in eukaryotic organisms, and the presence of the molecule is an indication for infection with pathogenic bacteria. Indeed, c-di-AMP is recognized by the human immune system. It binds the STING protein thus resulting the induction of an innate immune response [10].

The essentiality and toxicity of c-di-AMP for many bacteria have induced a great interest in understanding the reasons behind it. Classically, genes were regarded to be essential if they were absolutely needed for growth under optimal growth conditions, typically on complex medium at 37˚C [24–26]. However, many genes are not described as being essential because there may be paralogues that encode very similar functions as has been described for the c-di-AMP producing diadenylate cyclases in *B. subtilis* [8,9]. In some cases, one essential function can be carried out by very different proteins, and such genes are essential only under specific conditions, *i. e.* the absence of amino acids from the medium makes the corresponding

biosynthetic genes essential. These genes have been called conditional essential genes. Finally, some genes may not be essential but their loss would cause a substantial decrease of fitness for the cells, resulting either in the disappearance of such mutants or in the acquisition of suppressor mutations. Such genes or proteins have been designated quasi-essential, and the effect of their loss has been studied for the *B. subtilis* DNA topoisomerase TopA, the endoribonuclease RNase Y, and the modulator of lipid biosynthesis, YqhY [27–29]. Very recently, essential processes have been ranked on a global scale for the first time by the determination of the speed of death of mutants [30].

Two lines of research have provided important insights into the reasons of c-di-AMP essentiality. Studies to identify target molecules that interact with c-di-AMP demonstrated that many proteins involved in potassium and osmolyte homeostasis–both transporters and their regulators–are controlled by c-di-AMP to limit the intracellular concentrations of potassium and osmotically compatible compounds to physiologically reasonable levels [11,13,23,31–38]. In a complementary approach, suppressor mutants of bacterial strains that either lack or accumulate c-di-AMP have been studied. These analyses revealed that c-di-AMP is conditionally essential in *L. monocytogenes*, *S. aureus* and *B. subtilis*. While the nucleotide is required for growth on complex media, it is dispensable on minimal media [21,39,40]. For a *B. subtilis* mutant lacking c-di-AMP, growth was only possible on minimal medium as long as the potassium concentration in the medium is low and glutamate is not present [21,41]. On complex medium, the lack of c-di-AMP leads to an accumulation of (p)ppGpp [39], the second messenger that results in a global reprogramming of transcription, translation, and metabolism upon amino acid starvation. Inactivation of the enzymes that produce (p)ppGpp restores the viability of the strain lacking c-di-AMP on complex medium. The accumulation of (p)ppGpp leads to the inability of the transcription factor CodY to bind to its targets and thus in deregulation of a large regulon that includes many genes for osmolyte uptake and amino acid metabolism, and this uncontrolled expression of the CodY regulon is responsible for the essentiality of c-di-AMP for *L. monocytogenes* on complex media [39]. In *S. aureus*, c-di-AMP is dispensable for growth on complex medium under anaerobic conditions or when the bacteria acquire suppressor mutations that affect osmolyte or glutamine uptake [40,42]. Both types of studies support the idea that the control of potassium and osmolyte homeostasis is the central essential function of c-di-AMP in many bacteria that produce it [43–45].

We are interested in c-di-AMP signaling in the model organism *B. subtilis*. In contrast to *S. aureus* or *L. monocytogenes*, *B. subtilis* encodes three diadenylate cyclases that produce c-di-AMP, the constitutively expressed enzymes CdaA and DisA, and the sporulation-specific enzyme CdaS [9,46]. Mutants lacking both the *cdaA* and *disA* genes that encode the constitutive enzymes are not viable on complex medium and tolerate only very low potassium concentrations and no glutamate in minimal medium. In *B. subtilis*, c-di-AMP binds to two signal transduction proteins, DarA and DarB, to several potassium, osmolyte and magnesium transporters, as well as to the riboswitch that controls the expression of the high affinity potassium transporters KtrAB and KimA [20,23,47]. The DarA protein is a member of the large PII superfamily of signal transduction proteins that modulate the acitivities of transporters, enzymes, and transcription factors upon ligand binding [47,48]. However, the precise function of DarA has not yet been elucidated. DarB consists of two so-called CBS domains. Under conditions of potassium starvation, DarB is present as an apo-protein and then binds the (p)ppGpp synthetase/ hydrolase Rel to trigger an increase of the (p)ppGpp concentration, and thus to switch off cellular activities [49,50]. *B. subtilis*, encodes three potassium uptake systems, the paralogous KtrAB and KtrCD channels, and the KimA transporter as well as three potassium exporters, KhtTU, CpaA, and YugO [51]. With the exception of the sporulation protein

YugO, c-di-AMP binds to all proteins involved in potassium homeostasis, and inhibits and stimulates the activities of importers and exporters, respectively [23].

To get a better understanding of the global role of c-di-AMP for the physiology of *B. subtilis*, we performed extensive suppressor analysis of strains lacking c-di-AMP that had adapted to tolerate the presence of glutamate or that are viable on complex medium. These suppressor analyses were complemented by a transcriptome analysis to understand the roles of glutamate and potassium as well as of c-di-AMP for global gene expression in *B. subtilis*. Our results indicate that the adaptation to either glutamate or complex medium usually requires the simultaneous presence of multiple suppressor mutations that often affect potassium uptake, osmolyte uptake, or glutamate metabolism. Moreover, a mechanosensitive channel and phospholipid biosynthesis play an important role in the sensitivity of a strain lacking c-di-AMP to glutamate. Strikingly, this study identified the amino acid transporter YbeC (renamed AimA, <u>a</u>mino acid <u>i</u>mporter <u>A</u>) as the major glutamate transporter in *B. subtilis*. On complex medium, the c-di-AMP-binding signal transduction proteins DarA and DarB seem to be toxic in the absence of c-di-AMP, and only in their absence we could isolate suppressor mutants. The obtained results shed light on the complexicity of c-di-AMP essentiality and open new directions of research.

## Results

### Growth of a strain lacking c-di-AMP in the presence of glutamate requires distinct combinations of suppressor mutations

A *B. subtilis* strain lacking all three diadenylate cyclases is unable to grow at potassium concentrations of 5 mM or higher or in the presence of glutamate (even at low potassium concentrations) [21,41]. To test the sensitivity of the mutant for glutamate in more detail, we compared the growth of wild typ strains and the c-di-AMP-free strain GP2222 (designated Δ*dac* from here on) at increasing glutamate concentrations (see S1 Fig). While the wild type strain grew better in the presence of glutamate than with ammonium as the single source of nitrogen, the Δ*dac* mutant was inhibited already at very low glutamate concentrations (0.05%). In order to get more comprehensive insights into the toxicity of glutamate for the Δ*dac* mutant, we decided to isolate and study a larger set of suppressor mutants that tolerate the presence of glutamate. For this purpose, we plated the Δ*dac* strain [21] on MSSM medium containing glutamate (1% w/v) and a low concentration of potassium (0.1 mM KCl). While suppressor mutants appear readily after overnight cultivation when potassium toxicity was the growth-limiting problem [21], the toxicity of glutamate could only be overcome after several days. Moreover, the number of suppressor mutants was strongly reduced as compared to selection in the presence of potassium and ammonium as the nitrogen source. After four days at 37°C, 14 colonies could be isolated (see Fig 1, Δ*dac*). These strains were subjected to whole genome sequencing to identify the responsible mutations (Table 1). The long time for suppressor mutations to appear can be explained by each mutant harboring multiple mutations. In addition, this explains why the overall number of suppressors was rather low.

As observed previously [41], three strains were affected in *ktrC*. In one of these strains (GP3079), the genomic region from *abh* to *ktrC* was deleted resulting in reduced uptake of potassium via KtrCD. Similarly, strain GP3078 has a mutation affecting the high-affinity potassium transporter KtrB, and GP2842 also carries a mutation (*nhaK*) that affects potassium homeostasis. Interestingly, the same mutation in the NhaK cation:proton antiporter NhaK (S187F) had been isolated in a suppressor screen of the Δ*dac* strain in the presence of 5 mM potassium [21] indicating that the mutant protein has an increased activity in potassium export.

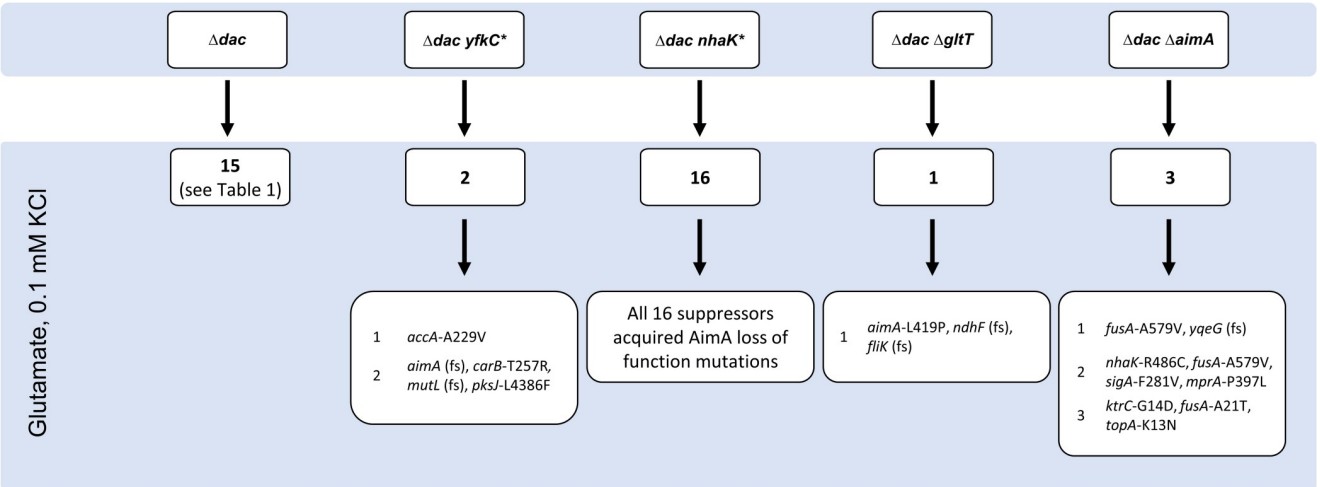

**Fig 1. A suppressor screen with a strain lacking c-di-AMP reveals the complexity of glutamate toxicity.** GP2222 (Δ*dac*), GP2814 (Δ*dac yfkC**), GP2223 (Δ*dac nhaK**), GP2259 (Δ*dac* Δ*gltT*), and GP3054 (Δ*dac* Δ*aimA*) were cultivated in MSSM medium with glutamate and 0.1 mM KCl and suppressors were isolated and analyzed by whole genome sequencing. Note that for GP2222 (Δ*dac*), 14 mutants were isolated in this study and one mutant was isolated previously [41] but studied in depth here, thus resulting in a total number of 15 mutants.

As mentioned above, the suppressor strains usually had multiple mutations. Since we never experienced the systematic acquisition of multiple mutations in prior suppressor screens [15,27,52–54], it seems likely that more than one mutation is required to achieve the suppression. Since the selective pressure in our experiment was directed towards tolerating otherwise toxic glutamate, we expected that the suppressor mutants would contain mutations affecting

**Table 1. Characterization of suppressor mutants derived from the Δ*dac* strain GP2222 in MSSM minimal medium with glutamate and 0.1 mM KCl[1].**

| Strain | Genotype | Ammonium, 0.1 mM KCl | Ammonium, 5 mM KCl | Glutamate, 0.1 mM KCl | Glutamate, 5 mM KCl | LB |
|---|---|---|---|---|---|---|
| GP2840[2] | *ktrC* (fs) *plsC*-A61V *ndhF* (fs) *sigD*-G88D | +++ | +++ | +++ | +++ | - |
| GP3079 | Δ(*abh-kinC-ykqA-ktrC*) *plsC*-L48P *yfkC*-R42W *fliP* (fs) *hbs*-I32T *ndoA*-G6S | ++ | +++ | +++ | +++ | ++ + |
| GP3077 | *ktrC* (fs) *aimA/glpQ* (trunc) *ybcC* (fs) | +++ | +++ | +++ | +++ | - |
| GP2841 | *gudB*+ *accC*-H298Y *fliG* (fs) | + | + | ++ | + | - |
| GP2842 | *nhaK*-S187F *aimA* (fs) | +++ | +++ | ++ | + | - |
| GP3078 | *plsC*-A61V *aimA*-T374M *ktrB*-G313D *sigD*-G88D | +++ | + | +++ | - | - |
| GP2849 | *ybcC*-K722L, I724F *aimA* (fs) | +++ | +++ | +++ | +++ | - |
| GP2837 | *plsC*-L48P *yfkC*-R42W *fliP* (fs) | +++ | +++ | +++ | +++ | ++ |
| GP3099 | *plsC*-L48P *yfkC*-R42W *fliP* (fs) *cotO*-I34F | +++ | + | +++ | - | - |
| GP3464 | *plsC*-P110S *flhA* (fs) | ++ | ++ | ++ | - | + |
| GP2224 | *yqeG* (fs) *aimA* (fs) *yqeV* (fs) | +++ | +++ | +++ | +++ | - |
| GP2838 | *ndhF*-V237E, M241L, H242L, A243D, G244C, L245F ΔICEBs1 (*int-yddM*) *yoaI* (fs) | +++ | ++ | ++ | ++ | + |
| GP3473 | *ndhF* (fs) *aimA* (fs) *gudB*+ *fliH* (fs) *yuiE*-G37V | ++ | ++ | ++ | - | - |
| GP3474 | *ndhF* (fs) *aimA* (fs) *ykpA* (fs) *ktrC* (fs) *fliK* (fs) | ++ | +++ | +++ | +++ | - |
| GP2839 | *fliK* (fs) *rpoB*-Q109P *rbsB*-R175L | ++ | ++ | ++ | - | - |

1 Growth on different media was judged after plating on corresponding agar plates. +++, colony growth as observed with the wild type strain; ++, growth slightly reduced as compare to the wild type strain; +, faint growth; -, no growth.

2 This strain was isolated in a previous study [41].

glutamate metabolism. Indeed, the á priori cryptic glutamate dehydrogenase was decryptified by the deletion of nine base pairs in two mutants (GP2841 and GP3473), thus allowing glutamate degradation [55,56]. Moreover, seven independently isolated mutants carried mutations affecting an amino acid transporter, YbeC. This protein has recently been identified as the major serine transporter of *B. subtilis* [57], and it is tempting to speculate that YbeC is the major glutamate transporter as well (see below). However, seven of the isolated mutants did not carry any mutation that is easily linked to glutamate homeostasis. The previously reported suppressor mutant GP2840 had mutations affecting PlsC, NdhF, and SigD (acyl-ACP:1-acyl-glycerolphosphate acyltransferase, NADH dehydrogenase, Sigma factor σ^D, respectively) in addition to the *ktrC* mutation [41]. Strikingly, *plsC* and *ndhF* are affected in six and four different suppressor mutants respectively. Several mutants carried mutations affecting motility: In addition to two *sigD* mutants, seven additional mutants carried mutations in flagellar genes that on one hand depend on a functional SigD for their expression, and on the other hand are required for SigD activity [58]. In addition, three of the 14 different independent suppressor mutants carry the same point mutation in the *yfkC* gene, encoding a mechanosensitive channel that releases ions from the cell [59,60]. This mutation results in a substitution of Arg-42 by a Trp residue.

We then tested the ability of the suppressor mutants to grow on minimal medium at low and high potassium concentrations with ammonium or glutamate as the nitrogen source. Moreover, the ability to grow on a complex medium was tested (see Table 1 and S2 Fig). Since the mutants were isolated in the presence of glutamate and 0.1 mM potassium, it is not surprising that all strains grew on the corresponding medium. Moreover, most of the strains were viable with ammonium as the single nitrogen source irrespective of the potassium concentration. Most of the mutants were also capable of growing at the elevated potassium concentration in the presence of glutamate. However, only four of the 15 tested strains were viable on complex medium. Two of them grew well on LB and these strains had the identical combination of mutations in *yfkC* and *plsC* (see Table 1, see below). This suggests that the adaptation to complex medium requires more than resistance to potassium and glutamate.

## Stepwise adaptation of a strain lacking c-di-AMP to growth on complex medium

Most of the suppressor mutants that tolerate the presence of glutamate and potassium were unable to grow on complex medium under standard laboratory conditions. The standard definition of essentiality refers to optimal growth conditions, *i. e.* to a complex medium and growth at 37˚C in the presence of oxygen for *B. subtilis* [25,26]. This prompted us to investigate the growth requirements of the Δ*dac* mutant GP2222 more rigorously. First, we attempted to isolate suppressor mutants derived from GP2222 under standard conditions as defined above. However, we never succeeded in isolating a single suppressor mutant. A recent study on c-di-AMP essentiality in *S. aureus* revealed that the nucleotide is dispensable under anaerobic conditions [40]. Therefore, we tried to isolate suppressor mutants of GP2222 in the absence of oxygen. While the wild type strain *B. subtilis* 168 grew under these conditions, again no suppressor mutants of the Δ*dac* strain could be isolated (see Fig 2).

In *B. subtilis*, c-di-AMP binds two signal transduction proteins, DarA and DarB [23,47]. In a suppressor screen using a *L. monocytogenes* strain lacking c-di-AMP, several mutations in the corresponding genes were observed [39]. We therefore constructed strains that lack both c-di-AMP and either the *darA* or the *darB* gene, and used these strains to isolate suppressors on complex medium under aerobic and anaerobic conditions. Again, no suppressors could be obtained under aerobic conditions; however, we detected one and two suppressors for the

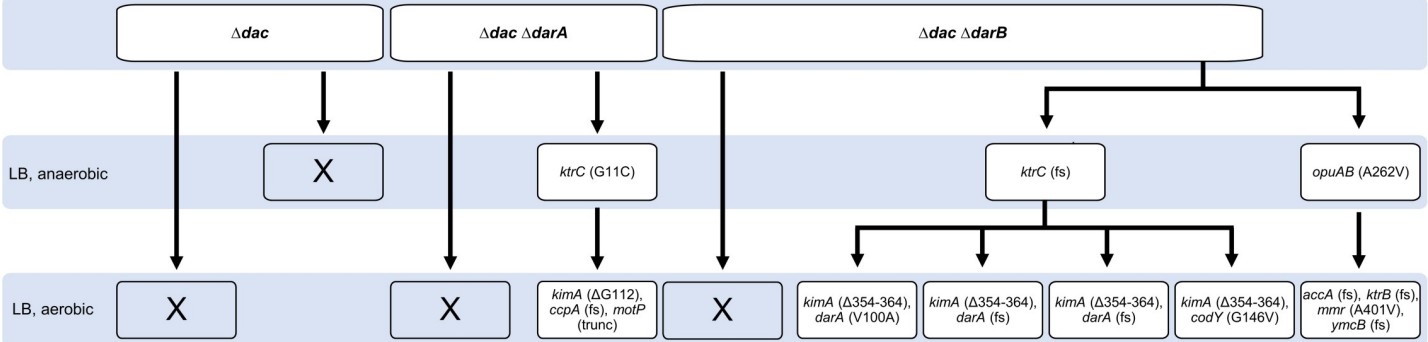

**Fig 2. A suppressor screen with a strain lacking c-di-AMP on complex medium.** GP2222 (Δ*dac*), GP2420 (Δ*dac* Δ*darA*), or GP2779 (Δ*dac* Δ*darB*) were cultivated on LB agar plates and incubated under aerobic or anaerobic conditions as indicated. The table shows how the parental strain evolved, and under which conditions suppressors could be isolated. The suppressor mutants were analyzed by whole genome sequencing.

Δ*dac* Δ*darA* and Δ*dac* Δ*darB* mutants, respectively, under anaerobic conditions (see Fig 2). Whole genome sequencing revealed that the mutation present in the Δ*dac* Δ*darA* mutant changed Gly-11 of KtrC to a cysteine residue. The suppressor mutants derived from the Δ*dac* Δ*darB* mutant had a frameshift in *ktrC* resulting in the expression of a truncated KtrC protein or a mutation affecting the permease domain OpuAB of the OpuA glycine betaine transporter [61], respectively (Fig 2). These results strongly support the idea that the accumulation of potassium or of osmoprotective compatible solutes is the reason for the essentiality of c-di-AMP [39,40,43].

An initial characterization of these suppressor mutants revealed that they were viable under anaerobic conditions, but not in the presence of oxygen. Therefore, we used the three strains for a further round of suppressor isolation. Indeed, we were able to obtain suppressors that tolerate the presence of oxygen for all three mutants (Figs 2 and S3). Whole-genome sequencing revealed that the initial suppressors with defective KtrC proteins had all acquired very specific mutations in the *kimA* gene, encoding the major high-affinity potassium transporter KimA. In the mutant isolated from the *darA ktrC* mutant, the absolutely conserved glycine in transmembrane helix 3 of KimA was deleted. Helix 3 is essential for potassium binding of KimA [62]. All four independent suppressors isolated from the Δ*dac* Δ*darB ktrC* mutant had a deletion of 11 amino acids in the transmembrane helix 9. Based on the high conservation of the deleted residues, we conclude that the mutant KimA proteins were inactive or less active than the wild type protein. Again, the *kimA* suppressor mutations were accompanied by other mutations (see Fig 2). In the suppressor mutant GP3010 derived from the Δ*dac* Δ*darA ktrC* mutant, we found a 155 bp deletion that results in the truncation the global transcription regulator for catabolic genes and operons, CcpA, and in the conversion of the start codon of *motP*, encoding the Na+-coupled MotP flagellar stator protein to a stop codon. For the Δ*dac* Δ*darB ktrC* suppressor mutant, two aerobic suppressors were subjected to whole genome sequencing. In one strain (GP2845), we found a point mutation in the *darA* gene that results in a substitution of the conserved Val-100 residue by an alanine in the DarA protein. The second suppressor (GP3094) carried mutations affecting the CodY transcription factor (G146V substitution) and the unknown SPβ phage protein YonD (D255H). In two additional suppressor mutants, we found identical frameshift mutations in *darA* that result in a truncation of the DarA protein after amino acid 64. Finally, we also characterized an aerobic suppressor (GP2847) derived from the anaerobic Δ*dac* Δ*darB opuAB* suppressor mutant. In this case, again a potassium channel subunit (KtrB) was inactivated as the result of a frameshift mutation. Moreover, this strain carried a frameshift mutation in the essential *accA* gene encoding a subunit of acetyl-CoA carboxylase that resulted in the formation of a truncation after amino acid 108 and

mutations in the genes encoding the putative methylenomycin A exporter Mmr and the tRNA methylthiotransferase YmcB. These results again highlight the importance of potassium homeostasis for the adaptation of strains lacking c-di-AMP.

Taken together, a *B. subtilis* strain lacking c-di-AMP is only able to grow under standard laboratory conditions if it acquires several suppressing mutations. The results of the two suppressor screens in the presence of glutamate and directly on complex medium demonstrate a distinct selective pressure under both conditions resulting in different sets of mutations that allow aerobic growth on complex medium. The importance of the initial loss of the c-di-AMP receptor proteins DarA and DarB for the direct adaptation to complex medium demonstrates that these proteins interfere with the growth of *B. subtilis* if they are present in the ligand-free apo-state. Indeed, apo-DarB was recently shown to trigger growth-inhibitory (p)ppGpp synthesis by the Rel protein [49,50]. For DarA, this is the first time that a phenotype for the corresponding mutant is reported. The repeated isolation of mutations affecting potassium uptake even in iterative cycles of suppressor screens highlights the severe toxicity of potassium for *B. subtilis* in the absence of c-di-AMP.

## A global transcription study of the regulatory effects of potassium, glutamate, and c-di-AMP

In order to obtain deeper insights into the physiological role of c-di-AMP, we decided to study the impact of a lack of this second messenger on global gene expression by a transcriptome analysis as a complementary approach to the suppressor analysis. Since c-di-AMP has been implicated in the interplay of potassium and glutamate homeostasis, we analysed the global gene expression for the wild type strain *B. subtilis* 168 at low (0.1 mM) and high (5 mM) potassium concentrations and in the presence of ammonium and glutamate as the nitrogen source (S1 Text). Moreover, we used the c-di-AMP free strain (Δ*dac*) GP2222 as well as the isogenic suppressor mutant GP2223 that is able to grow at 5 mM potassium (see Fig 3 for the regulatory network and S1 Table). Strikingly, expression of several genes involved in fermentation and respiration was strongly reduced in the absence of c-di-AMP (see S1 Table and S4A Fig for validation by quantitative RT-PCR). The *ldh-lctP* and *cydABCD* operons encoding lactate dehydrogenase and the lactate exporter as well as the terminal quinole oxidase, are repressed by the NADH-responsive transcription factor Rex [63,64]. The about 100-fold repression of these genes in the absence of c-di-AMP suggests that the NADH levels are reduced in the mutant. This conclusion is also supported by the increased expression (seven-fold) of the *ndhF-ybcC* operon (see S1 Table) which codes for a NADH dehydrogenase and thus also contributes to the oxidation of $NADH_2$ to NAD [65]. Importantly, the *ndhF* gene and its downstream gene *ybcC* were affected in several of the suppressor mutants suggesting that these mutations counteract the increased expression of the conserved *ndhF-ybcC* operon. In addition to genes of the Rex regulon, genes of the SigO regulon that are involved in the response to acid stress [66] and several competence genes were severely reduced in the Δ*dac* mutant GP2222. The strong reduction of competence gene expression in response to the absence of c-di-AMP suggests that this second messenger might be required for genetic competence and transformation in *B. subtilis*. To test this hypothesis, we attempted transformation of strain GP2222 using chromosomal DNA. No transformants were obtained, whereas the control strain 168 was efficiently transformable. Thus, the lack of c-di-AMP results in a loss of genetic competence as a result of reduced competence gene expression.

It has been shown before that the expression of the high affinity potassium uptake systems KtrAB and KimA is repressed in the presence of potassium via a c-di-AMP sensitive riboswitch [20–22]. In agreement with those results, these genes were most strongly repressed by potassium in our analysis in the wild type strain 168. The expression of the *kimA* transporter gene was 112- and 17-fold repressed at 5 mM potassium in the presence of glutamate or

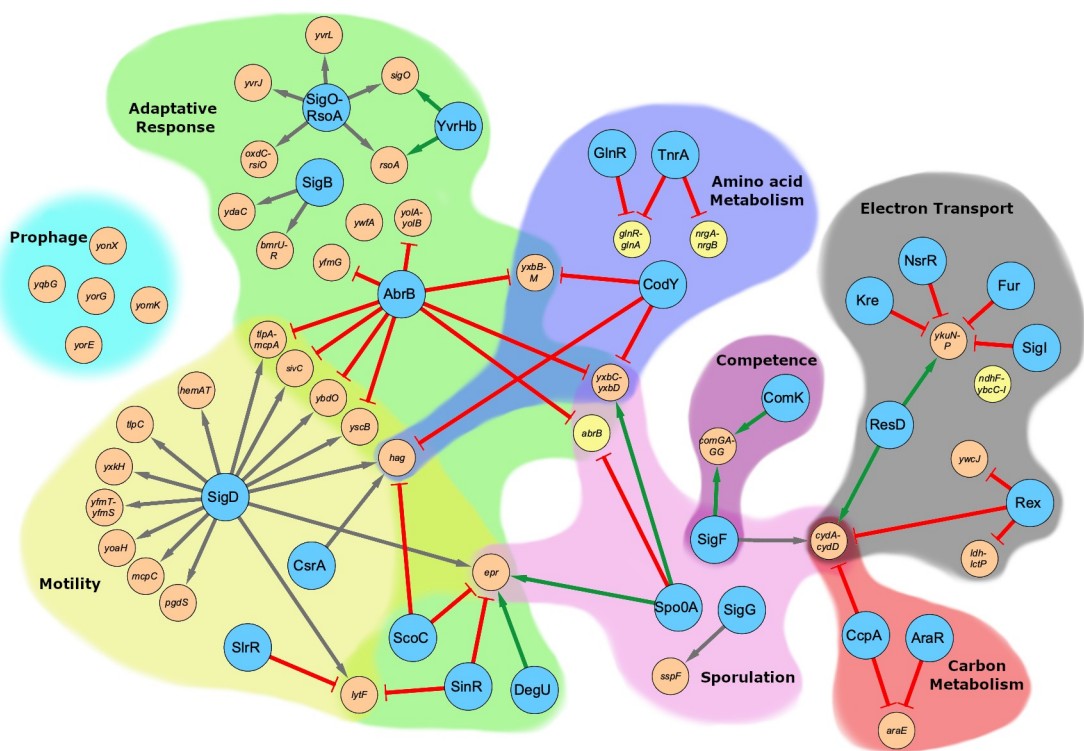

**Fig 3. Regulatory network of the most strongly up- and downregulated genes as a result of the lack of c-di-AMP.** Expression ratios of *B. subtilis* wild type and GP2222 (Δ*dac*) with ammonium and 0.1 mM KCl were calculated and the most up- or downregulated genes were clustered into operons, regulons, and grouped depending on their physiological function. Regulators are depicted as large blue nodes, and regulated operons or genes as yellow and orange nodes for up- and downregulated genes, respectively). Arrows: green: activation; gray: sigma-factor; red: repression.

ammonium, respectively; whereas the expression of the *ktrAB* channel operon was reduced 29- and 4-fold, respectively (see S2 Table). As expected, the expression of both *kimA* and *ktrAB* is increased in the absence of c-di-AMP; however, both genes are still repressed by potassium (eight- and five-fold, respectively), indicating a second, c-di-AMP independent mechanism of transcriptional regulation of these potassium transport systems (see S1 Table and S4B Fig for validation by quantitative RT-PCR, see S12 Fig for a reporter analysis). The particular importance of potassium for the cell and for KimA as a major high-affinity transporter is highlighted by the fact that the *kimA* gene belongs to the most strongly expressed genes at 0.1 mM potassium (position #50 in the presence of glutamate), and that it is even the most strongly expressed transporter in *B. subtilis* under these conditions (see https://www.ncbi.nlm.nih.gov/geo/query/acc.cgi?acc=GSE156738 for the full data set).

Taken together, the global transcription analysis supported the idea that c-di-AMP is involved in multiple cellular functions. Moreover, the analysis confirmed a particular role of genes involved in controlling NADH homeostasis, as already suggested by the suppressor analysis and by the differential effects of aerobic and anaerobic conditions to allow suppressor formation of the strains lacking c-di-AMP.

## A reduction of phospholipid biosynthesis helps to overcome the lack of c-di-AMP

The analysis of suppressor mutants that were viable in the presence of glutamate or on complex medium identified several mutations that affect proteins involved in phospholipid

biosynthesis, *i. e.* AccA, AccC, and PlsC (see Figs 1, 2 and S5 and Table 1 for an overview on lipid synthesis in *B. subtilis*). The *accA* and *accC* genes encode subunits of the essential acetyl-CoA carboxylase. Unexpectedly, a frameshift had occurred in the *accA* gene of strain GP2847. Since malonyl-CoA synthesis is the essential first step in fatty acid and phospholipid biosynthesis, it is likely that this mutation can be intrinsically suppressed and thus does not result in a complete loss of AccA synthesis. The *accA* mutation isolated in the Δ*dac yfkC** mutant results in an A229V substitution in the active site of the enzyme [67,68] (see S6 Fig). For AccC, the mutation (H298Y) is located in the highly conserved active site motif of the enzyme (S6 Fig). The localization of these mutations strongly suggests that the initial step of lipid biosynthesis, the formation of malonyl-CoA, is severely impaired in the suppressor mutants.

PlsC is the acyl-ACP:1-acylglycerolphosphate acyltransferase that catalyzes the formation of phosphatidic acid in the synthesis of phospholipids. Interestingly, the *plsC* gene is essential for the growth of *B. subtilis*. In agreement with this essentiality, none of the three different mutations in the six independent mutants results in a frame shift or a deletion. Instead, residue L48 is replaced by a proline in three independently isolated mutants, A61 by a valine in two mutants, and P110 by serine in one mutant. All three residues are located in functionally important regions of the protein. L48 is close to the HXXXXD motif, which is part the active site of the enzyme. Similarly, A61 and P110 are in very close proximity to the active center of the enzyme [69] (S7 Fig). The location of all three mutations in the immediate vicinity of the active center as well as the fact that the mutations affect conserved residues suggest that they result in a reduced PlsC activity, and thus in impaired phospholipid biosynthesis.

## The mechanosensitive channel YfkC contributes to the glutamate sensitivity of a c-di-AMP free strain

Three of the suppressor mutants isolated in the presence of glutamate carry mutations affecting the mechanosensitive channel of small conductance, YfkC (see Table 1). Strikingly, all three independently isolated mutants carry precisely the same mutation that results in a R42W substitution in the protein (see S8 Fig). This residue is located at the end of the first trans-membrane helix [70] and the mutation may affect the relative positioning of the trans-membrane helices 1 and 2. Another striking feature is that this mutation is in all three cases accompanied by point mutations in the *plsC* gene (see below). The presence of the same amino acid substitution in YfkC in all three strains suggested that this mutation resulted in a gain of function rather than in a loss of YfkC activity (from here on, we designate the mutant allele *yfkC**, the protein YfkC*). To address this question, we constructed Δ*dac* Δ*yfkC* and Δ*dac yfkC** strains lacking all diadenylate cyclases and assayed the growth of these strains on minimal medium containing glutamate or ammonium as the nitrogen source at high and low potassium concentrations (see Fig 4A). Both strains were viable when cultivated on plates containing ammonium and 0.1 mM potassium whereas they were unable to grow in the presence of glutamate or at the high potassium concentration. These observations indicate that neither the deletion of the *yfkC* gene nor *yfkC** mutation alone were sufficient to allow suppression of the glutamate and potassium toxicity of the Δ*dac* mutant. To address this issue further, we attempted the isolation of suppressor mutants from the Δ*dac* strain either lacking *yfkC* or carrying the *yfkC** allele in the presence of glutamate. No suppressor mutants could be isolated from the Δ*dac* Δ*yfkC* mutant, whereas two suppressor mutants were obtained from the Δ*dac yfkC** mutant (see Fig 1). This observation supports the idea that the specific R42W substitution of YfkC* is a prerequisite for the adaptation of the c-di-AMP-free strain to growth in the presence of glutamate. Whole genome sequencing of the two glutamate-resistant suppressor mutants identified a mutation in *accA* in strain GP2850 (see below). This mutation supports

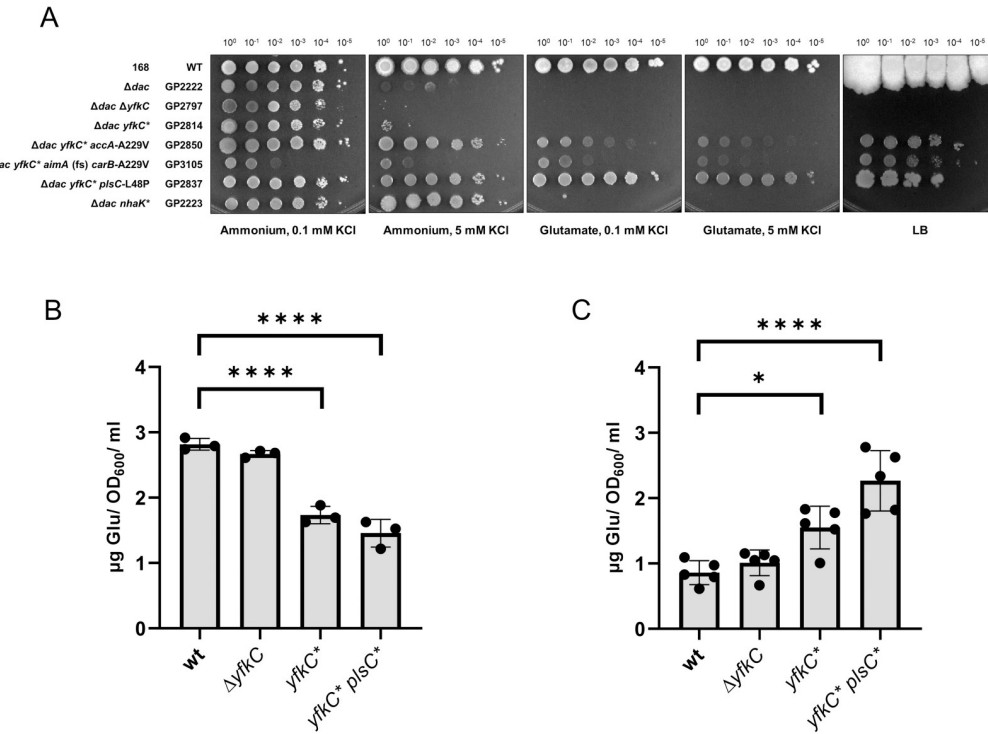

**Fig 4. The influence of the mechanosensitive channel YfkC on growth of a *B. subtilis* strain lacking c-di-AMP in the presence of glutamate.** (A) *B. subtilis* strains were cultivated in MSSM minimal medium with 0.1 mM KCl and glutamate. Cells were harvested, washed, and the $OD_{600}$ was adjusted to 1.0. Serial dilutions were dropped onto MSSM minimal plates with the indicated potassium concentration and ammonium or glutamate, or on LB plates. (B) The R42W amino acid substitution in the mechanosensitive channel YfkC reduces intracellular glutamate levels. The intracellular metabolites of *B. subtilis* wild type strain 168, and the isogenic Δ*yfkC*, *yfkC**, and *yfkC** *plsC** mutants were extracted and analyzed by GC-MS. (C) The R42W in the mechanosensitive channel YfkC results in an enhanced export of glutamate. The glutamate concentration in the supernatant of *B. subtilis* cultures of wild type strain 168, and the isogenic Δ*yfkC*, *yfkC**, and *yfkC** *plsC** mutants were analyzed by GC-MS. Statistical analysis was performed using a one-way ANOVA, followed by Tukey's multiple comparisons test (**** $P < 0.0001$). The numerical data are presented in S4 Table.

the requirement of paired mutations affecting *yfkC* and lipid biosynthesis. The second suppressor strain, GP3105, carries mutations affecting the amino acid transporter YbeC, the CarB subunit of the carbamoyl-phosphate synthetase, the endonuclease MutL and the polyketide synthase PksJ. Among these mutations, the frameshift in the *ybeC* gene is likely the most relevant (see below). These suppressor mutants were also tested for their growth in the presence of glutamate or increased potassium concentrations (see Fig 4A). Both were viable in the presence of glutamate irrespective of the potassium concentration. However, the growth in the presence of glutamate was much better in the case of the *yfkC** *plsC* (L48P) mutant GP2837 (Fig 4A). Taken together, these results suggest that the *yfkC** mutation in combination with the lipid biosynthetic or *ybeC* mutations is rather specific for the protection of the Δ*dac* mutant from glutamate toxicity. In the growth assay, we also included the Δ*dac nhaK* (S187F) mutant that has been isolated for resistance to potassium [21]. As expected, this strain grew at 5 mM potassium but was as sensitive to glutamate as the original Δ*dac* mutant, however we observed the formation of suppressor mutants in the presence of glutamate. As the NhaK allele used in this experiment is thought to exhibit increased potassium export activity [21], it is tempting to speculate the mutant mechanosensitive channel YfkC* is able to export glutamate to overcome the toxicity of this amino acid.

## The mutant forms of the mechanosensitive channel YfkC and the lipid biosynthetic enzyme PlsC allow glutamate export

The requirement of the $\Delta dac\ yfkC^*$ mutant for an additional mutation reducing lipid biosynthesis to allow growth in the presence of glutamate in several of the suppressor mutants suggests that the membrane properties are important for controlling cellular glutamate homeostasis. This conclusion is supported by the well-established functional interaction between mechanosensitive channels and lipid molecules [71,72]. It is tempting to speculate that the combination of altered lipid composition and the mutation in the trans-membrane helix of YfkC results in a change of channel properties that facilitates the export of glutamate by YfkC. To test this hypothesis, we determined the intracellular and extracellular glutamate concentrations in the wild type strain 168, and in the isogenic $\Delta yfkC$, $yfkC^*$ and $yfkC^*\ plsC^*$ mutants (Fig 4B and 4C). The wild type strain contained about 2.8 μg glutamate per $OD_{600}$ and ml, and the same glutamate concentration was detected in the $\Delta yfkC$ mutant. This is in good agreement with the observation that the deletion of the *yfkC* gene did not confer glutamate resistance to the $\Delta dac$ mutant. In contrast, the intracellular glutamate concentrations were reduced in the $yfkC^*$ mutant (1.7 μg/$OD_{600}$ and ml) and even more so in the $yfkC^*\ plsC^*$ double mutant (1.4 μg/$OD_{600}$ and ml). On the other hand, the glutamate concentration in the supernatant of the wild type strain and the $\Delta yfkC$ mutant was low (1 μg/$OD_{600}$ and ml), and a moderate increase to 1.3 μg/$OD_{600}$ and ml was observed in the presence of the $yfkC^*$ mutation. The extracellular glutamate concentration was most elevated in the $yfkC^*\ plsC^*$ double mutant (2.3 μg/$OD_{600}$ and ml). These results support the idea that the point mutation in YfkC* is crucial for lowering the intracellular glutamate concentration by facilitating export of the amino acid. In agreement with previous findings on the functional interactions between mechanosensitive channels and lipid molecules [71,72], the reduced lipid biosynthesis due to the PlsC* mutation further enhances glutamate export via YfkC*.

## AimA (YbeC) is a novel major glutamate transporter

The acquisition of suppressor mutations that provide resistance to glutamate to the $\Delta dac$ mutant was a rare event that took several days to occur (see above). However, we observed the rapid appearance of glutamate-resistant suppressor mutants if the strain already carried a *nhaK* mutation that facilitates potassium export. To determine the requirements for glutamate resistance in more detail, we isolated a set of 16 new suppressor mutants (see Fig 1). Since the originally isolated suppressor mutant GP2842 carried a combination of mutations in *nhaK* and the *ybeC* gene encoding an amino acid transporter, we sequenced the *ybeC* allele in all glutamate-resistant suppressor mutants isolated from the $\Delta dac\ nhaK$(S187F) (from now on referred to as *nhaK*\* and NhaK* for the gene and protein, respectively). Strikingly, all 16 strains carried mutations in *ybeC*. Among these mutations are frameshifts in ten strains, an insertion of 2 amino acids after residue 226 in one strain, and amino acid substitutions in the remaining five strains (for details, see Table 2). The high number of mutants with inactivated *ybeC* genes strongly suggests that the combination of NhaK* and loss of YbeC allows growth of the $\Delta dac$ mutant in the presence of glutamate. To test this hypothesis, we deleted the *ybeC* gene in the $\Delta dac$ mutant. The resulting strain, GP3054, was unable to grow in the presence of glutamate, indicating that only the combination of the *nhaK*\* and the *ybeC* mutations conferred resistance to glutamate to the $\Delta dac$ mutant.

Three lines of evidence suggest that YbeC is the major glutamate transporter of *B. subtilis*: (i) We isolated seven independent mutants affected in *ybeC* in the initial screen for mutations conferring glutamate resistance to the $\Delta dac$ strain, (ii) we reproducibly isolated a large number of *ybeC* mutants based on the $\Delta dac\ nhaK^*$ mutant, and (iii) YbeC is an amino acid transporter of the amino acid-polyamine-organocation (APC) superfamily. We have recently

**Table 2. Bacterial strains used in this study.**

| Strain | Genotype | Reference |
|---|---|---|
| 168 | *trpC2* | Laboratory collection |
| BKE14130 | *trpC2 ΔdarB::erm* | [90] |
| GP94 | *trpC2 ΔcdaA::spc* | [21] |
| GP342 | *trpC2 amyE::(gltA-lacZ kan)* | [91] |
| GP987 | *trpC2 ΔdisA::tet* | [9] |
| GP991 | *trpC2 ΔcdaS::ermC ΔdisA::tet* | [9] |
| GP997 | *trpC2 ΔcdaA::cat* | [9] |
| GP1360 | *trpC2 ΔcdaS::spc* | This study |
| GP1361 | *trpC2 ΔcdaS::spc disA::tet* | GP1360 → GP987 |
| GP1712 | *trpC2 ΔdarA::cat* | [47] |
| GP2181 | *trpC2 amyE::(P-kimA-RS-lacZ cat)* | pGP2912 → 168 |
| GP2222 | *trpC2 ΔcdaA::cat ΔcdaS::ermC ΔdisA::tet* | [21] |
| GP2223 | *trpC2 ΔcdaA::cat ΔcdaS::ermC ΔdisA::tet nhaK$_{S187F}$* | [21] |
| GP2224 | *trpC2 ΔcdaA::cat ΔcdaS::ermC ΔdisA::tet aimA $_{frameshift}$ yqeG$_{frameshift}$ yqeV$_{frameshift}$* | This study |
| GP2245 | *trpC2 ΔcdaS::ermC ΔdisA::tet ΔgltT::kan* | GP2248 → GP991 |
| GP2248 | *trpC2 ΔgltT::kan* | This study |
| GP2259 | *trpC2 ΔcdaA::cat ΔcdaS::ermC ΔgltT::kan ΔdisA::tet* | GP997 → GP2245 |
| GP2413 | *trpC2 ΔcdaS::ermC ΔdisA::tet ΔdarA::cat* | GP1712 → GP991 |
| GP2420 | *trpC2 ΔcdaS::ermC ΔdisA::tet ΔdarA::cat ΔcdaA::spc* | GP94 → GP2413 |
| GP2777 | *trpC2 ΔcdaS::spc ΔdisA::tet ΔdarB::ermC* | BKE14130 → GP1361 |
| GP2779 | *trpC2 ΔcdaS::spc ΔdisA::tet ΔdarB::ermC ΔcdaA::cat* | GP997 → GP2777 |
| GP2785 | *trpC2 ΔyfkC::kan* | This study |
| GP2786 | *trpC2 ΔaimA::kan* | This study |
| GP2796 | *trpC2 ΔcdaS::spc disA::tet ΔyfkC::kan* | GP2785 → GP1361 |
| GP2797 | *trpC2 ΔcdaS::spc disA::tet ΔyfkC::kan ΔcdaA::cat* | GP997 → GP2796 |
| GP2798 | *trpC2 yfkC$_{R42W}$ kan* | This study |
| GP2813 | *trpC2 ΔcdaS::spc disA::tet yfkC$_{R42W}$ kan* | GP2798 → GP1361 |
| GP2814 | *trpC2 ΔcdaS::spc disA::tet yfkC$_{R42W}$ kan ΔcdaA::cat* | GP997 → GP2813 |
| GP2818 | *trpC2 ΔgltP::cat ΔaimA::kan* | GP2786 → GP2824 |
| GP2824 | *trpC2 ΔgltP::cat* | This study |
| GP2825 | *trpC2 ΔgltT::kan ΔgltP::cat* | GP2824 → GP2248 |
| GP2831 | *trpC2 ΔgltT::kan ΔaimA::phleo* | LFH → GP2248 |
| GP2837 | *trpC2 ΔcdaA::cat ΔcdaS::ermC ΔdisA::tet yfkC$_{R42W}$ plsC$_{L48P}$ fliP$_{frameshift}$* | This study |
| GP2838 | *trpC2 ΔcdaA::cat ΔcdaS::ermC ΔdisA::tet ndhF$_{V237E, M241L, H242L, A243D, G244C, L245F}$ ΔydcL-yddM yoaI$_{frameshift}$* | This study |
| GP2839 | *trpC2 ΔcdaA::cat ΔcdaS::ermC ΔdisA::tet fliK$_{truncated}$ rpoB$_{Q109P}$ rbsB$_{R175L}$* | This study |
| GP2840 | *trpC2 ΔcdaA::cat ΔcdaS::ermC ΔdisA::tet plsC$_{A61V}$ ndhF$_{frameshift}$ ktrC$_{frameshift}$ sigD$_{G88D}$* | [41] |
| GP2841 | *trpC2 ΔcdaA::cat ΔcdaS::ermC ΔdisA::tet gudB$_{decrypt.}$ accC$_{H298Y}$* | This study |
| GP2842 | *trpC2 ΔcdaA::cat ΔcdaS::ermC ΔdisA::tet nhaK$_{S187F}$ aimA$_{frameshift}$* | This study |
| GP2845 | *trpC2 ΔcdaS::spc ΔdisA::tet ΔdarB::ermC ΔcdaA::cat ktrC (fs) kimA$_{Δ354-364}$ darA$_{V100A}$* | This study |
| GP2849 | *trpC2 ΔcdaA::cat ΔcdaS::ermC ΔdisA::tet aimA$_{frameshift}$ ybcC$_{K722L/I724F}$* | This study |

(*Continued*)

**Table 2.** (Continued)

| Strain | Genotype | Reference |
|---|---|---|
| GP2850 | *trpC2 ΔcdaS::spc disA::tet yfkC$_{R42W}$ kan ΔcdaA::cat accA$_{A229V}$ yerC$_{T58K}$ ypoP$_{KTEYF}$ $_{111\ I}$* | This study |
| GP3010 | *trpC2 ΔcdaS::ermC ΔdisA::tet ΔdarA::cat ΔcdaA::spc ykoX$_{K218STOP}$ ktrC$_{G11C}$ yfkF$_{G294S}$ nusG$_{T82K}$ kimA$_{ΔG112}$ ccpA$_{Δbp850-1005\ (incl\ 24bp\ downstream)}$* | This study |
| GP3053 | *trpC2 ΔcdaS::ermC ΔdisA::tet ΔaimA::phleo* | GP2831 → GP991 |
| GP3054 | *trpC2 ΔcdaS::ermC ΔdisA::tet ΔaimA::phleo ΔcdaA::cat* | GP997 → GP3053 |
| GP3071 | *trpC2 ΔgltT::kan ΔaimA::phleo ΔgltP::cat* | GP2824 → GP2831 |
| GP3077 | *trpC2 ΔcdaA::cat ΔcdaS::ermC ΔdisA::tet ktrC$_{frameshift}$ aimA/glpQ$_{truncation}$ ybcC$_{frameshift}$* | This study |
| GP3078 | *trpC2 ΔcdaA::cat ΔcdaS::ermC ΔdisA::tet plsC$_{A61V}$ aimA$_{T374M}$ ktrB$_{G313D}$ sigD$_{G88D}$* | This study |
| GP3079 | *trpC2 ΔcdaA::cat ΔcdaS::ermC ΔdisA::tet yfkC$_{R42W}$ plsC$_{L48P}$ Δ(abh-kinC-ykqA-ktrC) hbs$_{I32T}$ ndoA$_{G6S}$ fliP$_{frameshift}$* | This study |
| GP3093 | *trpC2 ΔcdaA::cat ΔcdaS::ermC ΔgltT::kan ΔdisA::tet aimA$_{L419P}$ ndhF$_{frameshift}$ fliK$_{frameshift}$* | This study |
| GP3094 | *trpC2 ΔcdaS::spc ΔdisA::tet ΔdarB::ermC ΔcdaA::cat ktrC$_{frameshift}$ kimA$_{Δ354-364}$ codY$_{G146V}$ yonD$_{D255H}$* | This study |
| GP3099 | *trpC2 ΔcdaA::cat ΔcdaS::ermC ΔdisA::tet plsC$_{L48P}$ yfkC$_{R42W}$ fliP$_{frameshift}$ cotO$_{I34F}$* | This study |
| GP3101 | *trpC2 ΔcdaS::ermC ΔdisA::tet ΔaimA::phleo ΔcdaA::cat fusA$_{A579V}$ yqeG$_{frameshift}$* | This study |
| GP3102 | *trpC2 ΔcdaS::ermC ΔdisA::tet ΔaimA::phleo ΔcdaA::cat nhaK$_{R486C}$ sigA$_{F281V}$ mprA$_{P397L}$ fusA$_{A579V}$* | This study |
| GP3103 | *trpC2 ΔcdaS::ermC ΔdisA::tet ΔaimA::phleo ΔcdaA::cat ktrC$_{G14D}$ topA$_{K13N}$ fusA$_{A21T}$* | This study |
| GP3105 | *trpC2 ΔcdaS::spc disA::tet yfkC$_{R42W}$ kan ΔcdaA::cat aimA$_{frameshift}$ carB$_{T257R}$ mutL$_{frameshift}$ pksJ$_{L4386F}$* | This study |
| GP3464 | *trpC2 ΔcdaA::cat ΔcdaS::ermC ΔdisA::tet plsC$_{P110S}$ flhA$_{frameshift}$* | This study |
| GP3473 | *trpC2 ΔcdaA::cat ΔcdaS::ermC ΔdisA::tet ndhF$_{frameshift}$ aimA$_{frameshift}$ gudB$^+$ fliH$_{frameshift}$ yuiE$_{G37V}$* | This study |
| GP3474 | *trpC2 ΔcdaA::cat ΔcdaS::ermC ΔdisA::tet ndhF (fs) aimA$_{frameshift}$ ykpA$_{frameshift}$ ktrC$_{frameshift}$ fliK$_{frameshift}$ fliH$_{silent}$* | This study |
| GP3486 | *trpC2 ΔcdaA::cat ΔcdaS::ermC ΔdisA::tet nhaK$_{S187F}$ aimA$_{frameshift}$* | This study |
| GP3487 | *trpC2 ΔcdaA::cat ΔcdaS::ermC ΔdisA::tet nhaK$_{S187F}$ aimA$_{frameshift}$* | This study |
| GP3488 | *trpC2 ΔcdaA::cat ΔcdaS::ermC ΔdisA::tet nhaK$_{S187F}$ aimA$_{frameshift}$* | This study |
| GP3489 | *trpC2 ΔcdaA::cat ΔcdaS::ermC ΔdisA::tet nhaK$_{S187F}$ aimA$_{P279H}$* | This study |
| GP3490 | *trpC2 ΔcdaA::cat ΔcdaS::ermC ΔdisA::tet nhaK$_{S187F}$ aimA$_{I164N}$* | This study |
| GP3491 | *trpC2 ΔcdaA::cat ΔcdaS::ermC ΔdisA::tet nhaK$_{S187F}$ aimA$_{frameshift}$* | This study |
| GP3492 | *trpC2 ΔcdaA::cat ΔcdaS::ermC ΔdisA::tet nhaK$_{S187F}$ aimA$_{frameshift}$* | This study |
| GP3493 | *trpC2 ΔcdaA::cat ΔcdaS::ermC ΔdisA::tet nhaK$_{S187F}$ aimA$_{frameshift}$* | This study |
| GP3494 | *trpC2 ΔcdaA::cat ΔcdaS::ermC ΔdisA::tet nhaK$_{S187F}$ aimA$_{S106P}$* | This study |
| GP3495 | *trpC2 ΔcdaA::cat ΔcdaS::ermC ΔdisA::tet nhaK$_{S187F}$ aimA$_{N349K}$* | This study |
| GP3496 | *trpC2 ΔcdaA::cat ΔcdaS::ermC ΔdisA::tet nhaK$_{S187F}$ aimA$_{frameshift}$* | This study |
| GP3497 | *trpC2 ΔcdaA::cat ΔcdaS::ermC ΔdisA::tet nhaK$_{S187F}$ aimA$_{frameshift}$* | This study |
| GP3498 | *trpC2 ΔcdaA::cat ΔcdaS::ermC ΔdisA::tet nhaK$_{S187F}$ aimA$_{G70R}$* | This study |
| GP3499 | *trpC2 ΔcdaA::cat ΔcdaS::ermC ΔdisA::tet nhaK$_{S187F}$ aimA$_{Insertion\ after\ aa\ 226\ (+VA)}$* | This study |
| GP3500 | *trpC2 ΔcdaA::cat ΔcdaS::ermC ΔdisA::tet nhaK$_{S187F}$ aimA$_{frameshift}$* | This study |
| GP3701 | *trpC2 ΔcdaA::cat ΔcdaS::ermC ΔdisA::tet nhaK$_{S187F}$ aimA$_{frameshift}$* | This study |
| GP3702 | *trpC2 ΔcdaS::spc ΔdisA::tet ΔdarB::ermC ΔcdaA::cat ktrC$_{frameshift}$ kimA$_{Δ354-364}$ darA$_{frameshift}$* | This study |
| GP3703 | *trpC2 ΔcdaS::spc ΔdisA::tet ΔdarB::ermC ΔcdaA::cat ktrC$_{frameshift}$ kimA$_{Δ354-364}$ darA$_{frameshift}$* | This study |

demonstrated that YbeC is the major serine transporter of *B. subtilis* [57]. It is well established that many amino acid transporters can take up multiple different amino acids. To test the possible role of YbeC in glutamate uptake, we first attempted functional complementation in *E. coli* that is unable to grow with glutamate as the single source of carbon and nitrogen due to the low activity of intrinsic glutamate transporters [73]. For this purpose, we transformed *E. coli* JM109 with plasmid pGP2987 that carries the *ybeC* gene or with the empty vector pWH844 [74]. The resulting transformants were tested for growth on minimal medium with glucose and ammonium or with glutamate as the only source of carbon and nitrogen (see Fig 5A). Whereas both strains grew well with glucose and ammonium, the strain carrying the empty vector was unable to utilize glutamate. In contrast, the expression of the *ybeC* gene allowed *E. coli* to use glutamate as source of carbon and energy. These findings confirm that YbeC indeed is a glutamate transporter. The protein and the gene were therefore renamed AimA and *aimA*, respectively (amino acid importer A).

It is very striking that our suppressor screens for glutamate-resistant mutants often identified AimA but never GltT or GltP, the two described glutamate transporters of *B. subtilis*. Therefore, we analyzed the contributions of these proteins to glutamate uptake by comparing the growth of strains expressing only one of the three proteins and a strain lacking all of them (see Fig 5B). All tested strains grew well in the presence of ammonium as the nitrogen source, reflecting their ability to synthesize glutamate. The strain GP3071 lacking all three presumptive glutamate transporters was only viable at high glutamate concentrations but not at 1 mM or 5 mM. This demonstrates that the bacteria need to transport glutamate if no ammonium is available for its synthesis. Moreover, it demonstrates that efficient glutamate transport is not possible in the absence of the three proteins. GP2818 expressing GltT grew at the three tested glutamate concentrations confirming its role as glutamate transporter. The strain GP2825 that expressed AimA as the only glutamate transporter did not grow at the lowest tested glutamate concentration (1 mM), but this strain was able to grow at 5 mM and 20 mM glutamate suggesting that AimA might be a low-affinity glutamate transporter. In contrast, the strain expressing GltP was only viable at highest glutamate concentration as observed for strain GP3071 lacking all three proteins. This suggests that GltP plays only a very minor, if any role, in glutamate uptake, and that yet another protein can transport glutamate at high concentrations.

To study the activities of AimA and GltT in glutamate transport in more detail, we assayed the growth of strains expressing only one of these transporters at different glutamate concentrations (see Fig 5C and 5D). Moreover, we performed this experiment in the presence of 0.1 mM and 5 mM KCl in the medium to see whether potassium has an effect on glutamate uptake as it has been observed with an activating effect of glutamate on potassium uptake by KtrCD [41]. As control, we used strain GP3071 that lacks AimA, GltT, and GltP. The results were used to calculate apparent $K_S$ values that reflect the affinities of the transporters for glutamate. For GltT, we observed $K_S$ values of 0.3 mM and 0.06 mM in the presence of 0.1 and 5 mM potassium, respectively. These results are in good agreement with the description of GltT as a high-affinity glutamate transporter [75]. Moreover, they show that the affinity of GltT for glutamate is five-fold increased in the presence of high potassium concentrations. For AimA, we determined apparent $K_S$ values of 3.2 and 2.3 mM in the presence of 0.1 and 5 mM potassium, respectively. In good agreement with the drop dilution assay, this suggests that AimA is a low-affinity glutamate transporter, and that the affinity of AimA is not affected by the external potassium concentration. The strain GP3071 lacking all known and suspected glutamate transporters grew only at high glutamate concentrations. This suggests the presence of an additional transporter with very low affinity for glutamate (apparent $K_S$ values of 12.3 and 13.8 mM in the presence of 0.1 and 5 mM potassium, respectively). Thus, AimA is the major glutamate transporter under standard growth conditions when glutamate is not limiting in the medium.

The role of AimA as the major glutamate transporter is in excellent agreement with the common isolation of glutamate-resistant Δ*dac aimA* mutants whereas no *gltT* mutants could be isolated. Indeed, the Δ*dac* Δ*gltT* mutant was unable to grow in the presence of glutamate, but again a suppressor mutant, GP3093, could be isolated (see Fig 1). Not surprisingly, this strain carries a mutation in *aimA* (Leu-419 Pro). Moreover, this strain carries frame-shift mutations in *ndhF* and *fliK*. The paired occurrence of *aimA* and *ndhF* suppressor mutations has been observed several times independently (see Table 1).

To get further insights into *gltP*, *gltT* and *aimA*, we determined the expression of these genes based on the transcriptome analysis under the relevant conditions (see Fig 5E). The expression of *gltP* was very low under all tested conditions, which is in agreement with the observation that GltP does not play a significant role in glutamate uptake. The expression of both GltT and AimA was significant. In the presence of glutamate in the medium, the expression was reduced five- and two-fold for AimA and GltT, respectively, at the high potassium concentration (5 mM). In the absence of glutamate, *aimA* expression did not respond to the supply of potassium whereas the expression of *gltT* was about two-fold increased at the increased potassium concentration. Thus, if glutamate is limiting, both the expression of *gltT* and the affinity of the GltT protein for glutamate are increased by potassium.

## Adaptation of a Δ*dac* Δ*aimA* mutant to growth in the presence of glutamate

To make the picture of the requirements for a Δ*dac* mutant to tolerate glutamate complete, we also adapted a Δ*dac* Δ*aimA* mutant to the presence of glutamate. As described above, this mutant was not viable in the presence of glutamate; however, we could isolate three glutamate-resistant suppressor mutants (see Fig 1). The mutations in these strains were identified by whole genome sequencing. Interestingly, all three suppressor mutants had mutations affecting the essential translation elongation factor G, in two strains (GP3101 and GP3103) the highly conserved Ala-579 was replaced by a Val (see S9 Fig). Ala-579 is located in immediate vicinity of a loop (Loop II) that is required for tRNA translocation in the ribosome [76,77], suggesting that the mutation interferes with translation. In the third strain (GP3102) we found an A21T substitution. This residue is the immediate neighbour of Asp-20, which binds the $Mg^{2+}$ ion in the active site of the protein [78]. The presence of substitutions in highly conserved and functionally important regions of FusA in all three suppressors suggests that reduced translation efficiency helps the Δ*dac* Δ*aimA* mutant to adapt to the presence of glutamate. In addition to the mutations in *fusA* each strain had a distinct additional mutation. Strain GP3101 had a frameshift mutation in the *yqeG* gene encoding a putative HAD superfamily phosphatase [79], whereas the two other strains carried point mutations affecting potassium homeostasis (*nhaK* and *ktrC* for GP3102 and GP3103, respectively) again highlighting the toxicity of potassium in the presence of glutamate. Moreover, GP3102 carried amino acid substitutions affecting SigA, the housekeeping sigma factor (F281V) and MrpA (P397L), a subunit of the sodium exporter of *B. subtilis*, and GP3103 had a point mutation affecting the quasi-essential DNA topoisomerase 1 TopA (K13N). The analysis of these strains for growth in the presence of glutamate with 0.1 or 5 mM potassium revealed that all suppressors were viable at both potassium concentrations, however, the suppression was most efficient if the *yqeG* or *nhaK* genes were affected in addition to *aimA* (see S10 Fig).

## Discussion

Cyclic di-AMP is essential f or many bacteria that produce this second messenger; however, the reasons behind this essentiality have not been fully understood. In this study, we used the

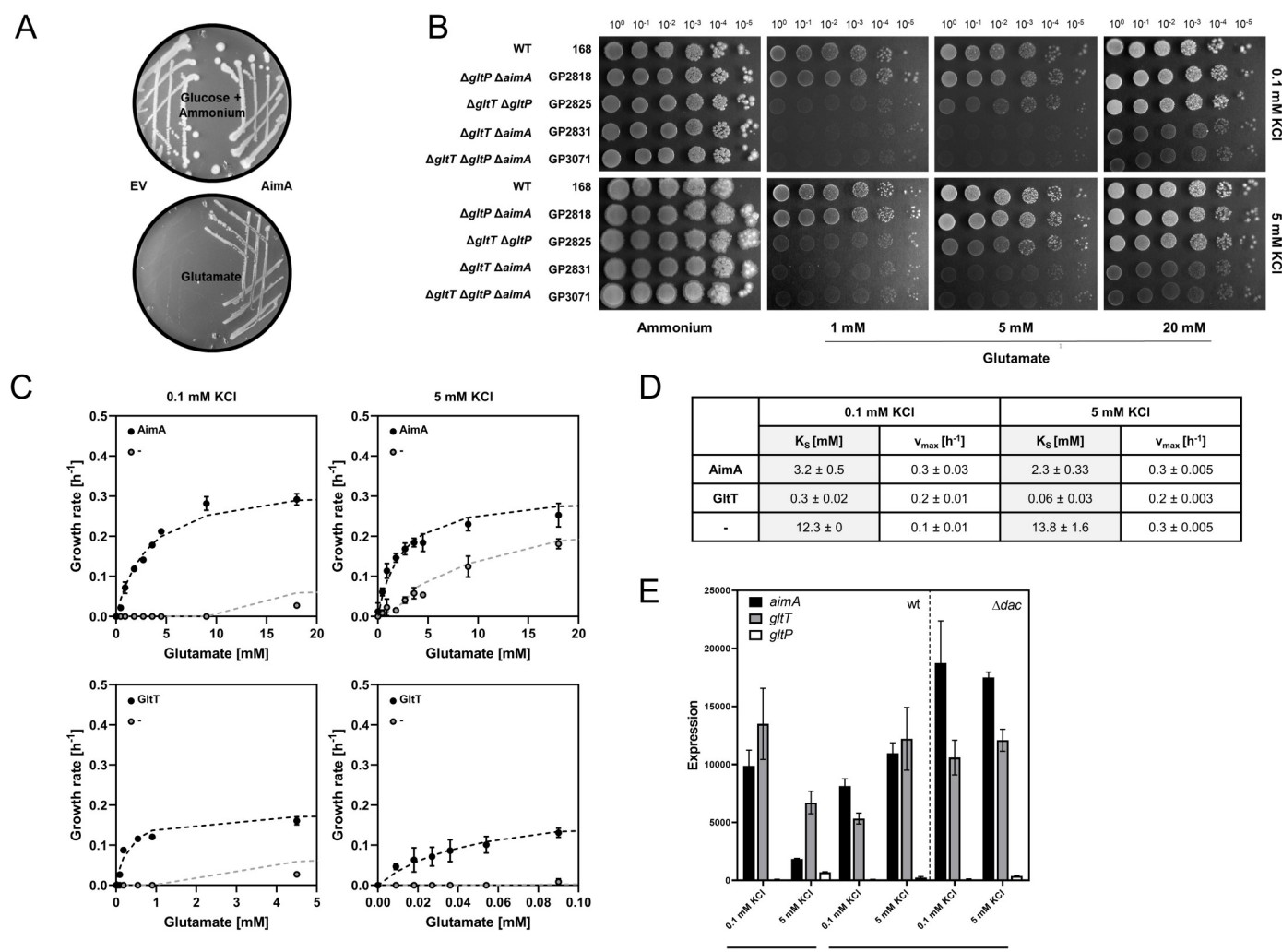

**Fig 5. Identification of AimA as a novel low-affinity glutamate transporter.** (A) *E. coli* JM109 was transformed with pGP2987 (AimA) and pWH844 (empty vector) and growth of *E. coli* JM109 was assessed on MSSM minimal medium with glucose and ammonium or glutamate as carbon and nitrogen source. (B) Growth assay of *B. subtilis* glutamate transporter mutants. *B. subtilis* strains were cultivated in MSSM minimal medium with 0.1 mM KCl and ammonium. Cells were harvested, washed, and the $OD_{600}$ was adjusted to 1.0. Serial dilutions were dropped onto MSSM minimal plates with the indicated potassium concentration and ammonium or glutamate (1, 5, or 20 mM). (C) Determination of glutamate transporter kinetics. The growth rates of *B. subtilis* strains expressing either AimA (upper panel) or GltT (lower panel) as the only glutamate transporter were used to determine the apparent glutamate transporter kinetics of AimA and GltT. A *B. subtilis* strain lacking all glutamate transporters (Δ*gltT* Δ*aimA* Δ*gltP*; light grey) served as a control. The strains were grown in MSSM minimal medium with 0.1 mM or 5 mM KCl and various glutamate concentrations. The growth rate was plotted against the glutamate concentration. (D) Kinetic parameters ($K_S$ and $v_{max}$ values) for AimA and GltT. The numerical data are presented in S4 Table. (E) Expression of *gltT*, *aimA*, and *gltP*. The expression levels of the genes encoding the glutamate transporters GltT, AimA, and GltP were extracted from the transcriptomic analysis.

Gram-positive model organism *B. subtilis* to gain further insights into the critical functions of c-di-AMP. Based on a variety of distinct suppressor screens to adapt a *B. subtilis* strain lacking c-di-AMP to either the presence of glutamate or to growth on complex medium, we can conclude that the control of potassium homeostasis is the major bottleneck that limits growth of the mutant under both conditions. This is in good agreement with (i) the large variety of c-di-AMP target proteins and RNA molecules that are involved in the uptake and export of potassium in *B. subtilis* and other bacteria [11,13,20,21,23,31,34,45,80], (ii) the fact that the intracellular c-di-AMP levels seem to report the extracellular potassium concentrations [18,21,37], and the isolation of suppressor mutants affecting potassium homeostasis in response to altered

cellular c-di-AMP levels also in other bacteria [18,37]. However, our study also clearly demonstrates that potassium is not the only problem for growth of *B. subtilis* in the absence of c-di-AMP since the presence of mutations that reduce potassium uptake or facilitate its export is not sufficient to overcome the toxicity of glutamate or complex medium. Suppressor studies with *L. monocytogenes* and *S. aureus* mutants lacking c-di-AMP revealed that those mutants often had acquired mutations in transporters for osmoprotectants such as glycine betaine indicating that intoxication by compatible solutes is the main reason for essentiality of c-di-AMP in these latter bacteria. The idea of a major role for c-di-AMP in osmoprotection rather than in potassium homeostasis in these pathogenic bacteria is also supported by the observation that potassium has only a minor impact on a *L. monocytogenes* strain lacking the only diadenylate cyclase [80]. Thus, while general features of c-di-AMP signaling are conserved in different bacteria, the specific roles and mechanisms may differ even between closely related bacteria.

Potassium and glutamate are the most abundant cation and anion, respectively, in every living cell, and the intracellular concentrations of the two ions need to be balanced [81]. In *B. subtilis*, c-di-AMP seems to play an important role in achieving this balance. The concentrations of the nucleotide respond both to the potassium and the nitrogen source availability, they are high at high potassium concentrations and in the presence of glutamate but not ammonium or glutamine [15,21,40]. Our work supports the idea of a close physiological integration of potassium and glutamate homeostasis: Both potassium and glutamate inhibit growth of the Δ*dac* mutant independently of each other. The accumulation of amino acids has also been shown to be a problem for the Δ*dac* mutants of *L. monocytogenes* and *S. aureus*. As presented in this study, in both organisms amino acid transporters were often inactivated in suppressor mutants that were capable of growing on complex medium despite the lack of c-di-AMP: In *L. monocytogenes*, several suppressor mutants carried mutations inactivating the OppABCD oligopeptide transporter [39] whereas the characterization of *S. aureus* suppressor mutants resulted in the identification of the AlsT glutamine transporter, thus indicating that glutamine rather than glutamate is toxic for the *S. aureus* Δ*dac* mutant [39,42]. Again, these differences between the related organisms support the idea that c-di-AMP has global overarching functions in the different bacteria, but these can be put into practice differently.

It had been established previously that the accumulation of glutamate can be toxic for *B. subtilis* [56,82]. In the case of the Δ*dac* mutant, our work has identified two essential requirements, i. e. the reduction of the intracellular glutamate and potassium concentrations (see Fig 6 for an overview). Since glutamate stimulates potassium uptake by the KtrCD channel [41] we can conclude that the increased potaasium uptake is the major reason for the toxicity of glutamate for the Δ*dac* mutant. Our work has unraveled three ways to reduce the intracellular glutamate concentrations: (i) Glutamate can be efficiently degraded upon decryptification of the otherwise inactive *gudB* gene encoding the major constitutive glutamate dehydrogenase of *B. subtilis*. Such a well established *gudB*+ mutation [55] was found in the suppressor mutants GP2841 and GP3473. (ii) The uptake of glutamate can be prevented, and this was indeed observed in many suppressor mutants. Surprisingly, the recently characterized serine transporter AimA (YbeC) was identified as the main glutamate transporter rather than the previsously studied GltT [75]. Even the Δ*dac* mutant lacking *gltT* was unable to grow in the presence of glutamate unless AimA had been inactivated (see Fig 1). This observation as well as our detailed kinetic analyses revealed that AimA is the major low affinity transporter for glutamate. Moreover, in the absence of c-di-AMP, the *aimA* gene is the most strongly expressed gene encoding a glutamate transporter (see Fig 5E). It is interesting to note that the expression of *aimA* responds to the availability of c-di-AMP (see Fig 5B), however, the mechanism responsible for this control has yet to be uncovered. We have previously demonstrated that a potassium channel (KtrCD) is directly activated and thus converted from a low- to a

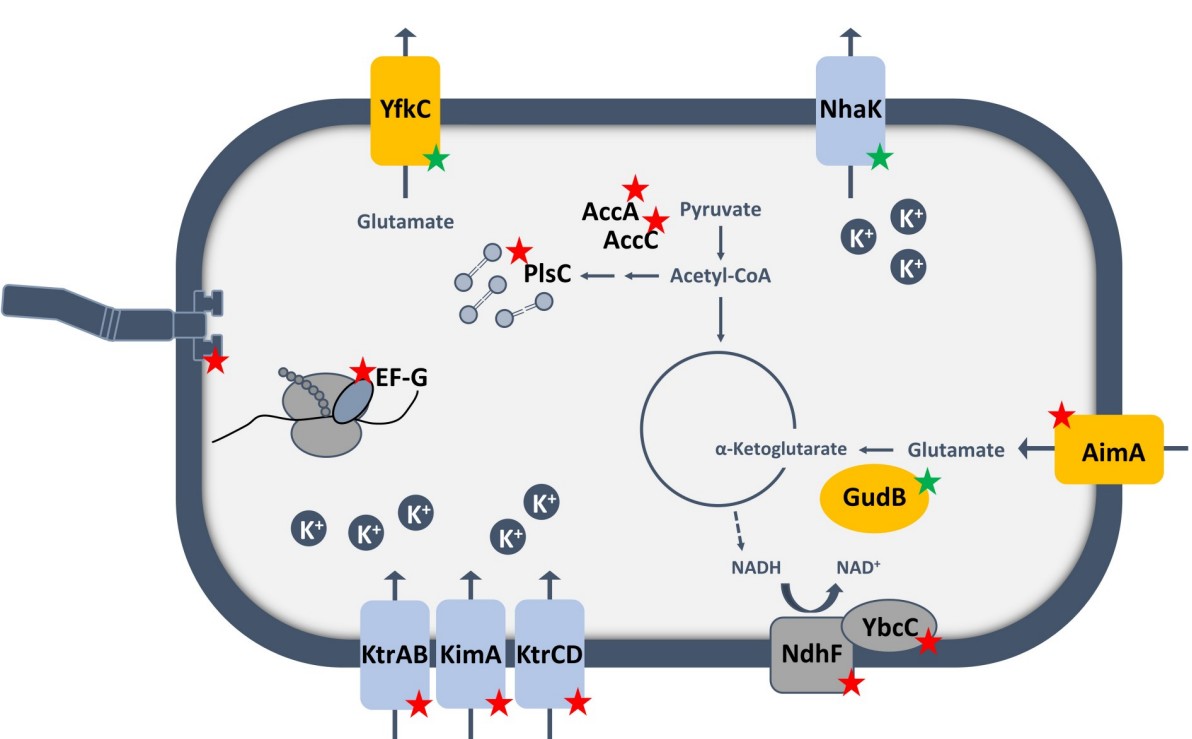

**Fig 6. Acquisition of glutamate resistence in a *B. subtilis* strain lacking c-di-AMP.** The pathways and functions affected in the isolated mutants obtained in the suppressor screen with glutamate (Table 1) and LB (Fig 2), are depicted in this model. Yellow: mutated proteins related to glutamate metabolism, blue: proteins involved in potassium homeostasis. Red stars indicate loss of function mutations, green stars indicate gain of function mutations.

high-affinity uptake system in the presence of glutamate [41]. Here, we show that similarly the affinity of the high affinity glutamate transporter GltT is controlled by the presence of potassium: At high potassium concentrations, the affinity of GltT for glutamate is increased fivefold. Thus, the uptake systems for the two most abundant ions in the cell can respond to the presence of the other ion to keep their intracellular concentrations balanced. (iii) In addition to mutations activating glutamate degradation or preventing glutamate uptake, we found several mutants with paired mutations affecting lipid biosynthesis and the mechanosensitive channel of small conductivity YfkC. Moreover, the Δ*dac yfkC** mutant expressing the altered channel protein had to acquire additional mutations affecting either glutamate uptake (*aimA*) or lipid biosynthesis (*accA*) to tolerate the otherwise toxic glutamate. We suggest that the altered YfkC* protein in conjunction with altered membrane properties facilitates the export of glutamate. This idea is supported by the observation that YfkC needs a gain-of-function mutation to bypass glutamate toxicity. Moreover, MscS-like mechanosensitive channels have been shown to export glutamate in *Corynebacterium glutamicum* [83,84]. In this bacterium, a gain of function mutation affecting this channel results in constitutive glutamate secretion [85]. Finally, specific membrane properties in *C. glutamicum* such as reduced lipid amounts facilitate glutamate export by the MscS-like channel (see [86] for review). Our data that demonstrate reduced intracellular and increased extracellular glutamate concentrations in the *yfkC** *plsC** double mutant provides strong evidence for the idea that the combination of mutations reducing lipid biosynthesis and that affect the mechanosensitive channel YfkC allow glutamate export in *B. subtilis*, thus overcoming the toxic effect of this amino acid.

Three lines of evidence suggest that respiration is a major bottleneck for *B. subtilis* in the absence of c-di-AMP. First, many suppressors isolated in the presence of glutamate had

mutations that inactivate the *ndhF* gene encoding a subunit of a putative NADH dehydrogenase or the downstream *ybcC* gene of unknown function. It is tempting to speculate that the two proteins form a complex that catalyzes the oxidation of $NADH_2$. Second, when we used complex medium to isolate suppressor mutants, this was initially only possible in the absence of oxygen if respiration activity was low. This is paralleled by the observation that the *S. aureus* Δ*dac* is viable in the absence but not in the presence of oxygen [40]. Finally, the lack of c-di-AMP results in an enhanced expression of the *ndhF-ybcC* operon and reduced intracellular $NADH_2$ levels as indicated by the strong repression of the genes of the Rex regulon in the Δ*dac* mutant. Although it is not yet understood how precisely respiration interferes with the growth of the Δ*dac* mutant, it is tempting to speculate that the proton motive force generated by respiration plays a key role: the major potassium transporter KimA is actually a proton:potassium symporter [62],and potassium uptake by the KtrAB and KtrCD channels is directly driven by the proton motive force [87]. Thus, a reduced respiration lowers the proton motive force and may thus limit the uptake of potassium, the major growth-limiting factor for the *B. subtilis* Δ*dac* mutant.

The adaptation of the Δ*dac* mutant to complex medium confirmed the toxicity of potassium. Surprisingly, none of the suppressors isolated on complex medium carried mutations specifically related to glutamate. The acquisition of suppressor mutations that allow growth under anaerobic conditions on complex medium was only possible when one of the c-di-AMP binding signal transduction proteins DarA or DarB was absent (see Fig 2). This indicates that these proteins inhibit growth in their c-di-AMP-free apo form. It is possible that DarA and/or DarB have functions related to glutamate metabolism, and that the deletions of either of the corresponding genes already covers the aspect of glutamate homeostasis. For DarB it has very recently been shown that the apo protein stimulates the synthesis of (p)ppGpp by the alarmone synthetase Rel both in *B. subtilis* and *L. monocytogenes* [49,50], and excessive (p)ppGpp synthesis limits growth of both organisms [39,49,50]. For DarA, no function has been identified so far. It is interesting to note, however, that DarA seems to be even more inhibitory than DarB since the Δ*dac* Δ*darB* mutants tend to acquire additional mutations affecting DarA, suggesting that DarA might be involved in the control of glutamate metabolism.

Taken together, our study supports the idea that c-di-AMP plays a central role in the physiology of *B. subtilis*. Moreover, by comparing the results obtained with suppressor screens in closely related but distinct bacteria, it becomes clear that the overall themes of c-di-AMP signaling are similar in different bacteria whereas the precise mechanisms are not. While some of the functions of c-di-AMP in *B. subtilis* such as the control of potassium homeostasis are already well understood, there remain many open questions for future research such as the link between c-di-AMP and lipid biosynthesis and the function of the PII-like signal transduction protein DarA. The work described here provides a solid ground for further investigation.

## Materials and methods

### Strains, media and growth conditions

*E. coli* DH5α [88] was used for cloning and for the expression of recombinant proteins. *E. coli* JM109 (Promega) was used for glutamate transporter complementation assays. All *B. subtilis* strains used in this study are derivatives of the laboratory strain 168. They are listed in Table 2. *B. subtilis* was grown in Luria-Bertani (LB) or in sporulation (SP) medium [88,89]. For the assay of potassium and glutamate toxicity, *B. subtilis* was cultivated in MSSM medium [21]. In this modified SM medium $KH_2PO_4$ was replaced by $NaH_2PO_4$ and KCl was added as indicated. The media were supplemented with ampicillin (100 μg/ ml), kanamycin (10 and 50 μg/ ml for *B. subtilis* and *E. coli*, respectively), chloramphenicol (5 μg/ ml), tetracyclin (12.5 μg/

ml), spectinomycin (150 μg/ ml) or erythromycin and lincomycin (2 and 25 μg/ ml, respectively) if required.

## Suppressor screen

For identification of suppressor mutations that rescue the growth of the c-di-AMP lacking strain GP2222, a suppressor screen was set up. The same set-up was performed for the suppressor screens with GP2223, GP2814, GP3054, and GP2259 (see Fig 1). The respective strain was plated on MSSM minimal plates in the presence of ammonium and 0.1 mM KCl. After incubation for 2–3 days, cell material was scraped from the plates and washed three times in MSSM basal salts solution. These cells were either plated on MSSM minimal plates (Na-Glutamate, 0.1 mM KCl) or used for inoculation of 10 ml MSSM minimal medium (Na-Glutamate, 0.1 mM KCl). The bacteria were then incubated at 42˚C for 3–4 days. As soon as suppressors appeared on the plates, they were picked and re-streaked three times. For liquid cultures, as far as growth could be observed, they were plated. Suppressors were picked and re-isolated again. For the suppressor screen on complex medium, GP2222, GP2420, and GP2779 were plated on LB plates. The plates were incubated at 37˚C either under standard conditions (aerobic) or in an anaerobic jar to create an anaerobic environment. As soon as suppressors appeared, they were picked and re-isolated three times. For the two step adaptation to complex medium, suppressor mutants previously isolated in the anaerobic screen were plated on LB plates and incubated at 37˚C under aerobic conditions. As soon as new suppressors appeared, they were picked and re-isolated three times. All isolated mutants were analyzed by whole-genome sequencing (see below).

## Phenotypic characterization

Amylase activity was detected on plates containing nutrient broth (7.5 g/liter), 17 g of Bacto agar per liter (Difco), and 5 g of hydrolyzed starch per liter (Connaught). The ability to degrade starch was detected by sublimating iodine onto the plates. For quantitative determination of *lacZ* expression in *B. subtilis*, cells were grown in MSSM medium with the nitrogen source and potassium concentration as indicated and harvested in the exponential phase. β-Galactosidase activity was determined with cell extracts obtained by lysozyme treatment as described previously [89]. One unit of β-galactosidase is defined as the amount of enzyme that produces 1 nmol of o-nitrophenol per minute at 28˚C. Experiments were performed in triplicate.

To assay growth of *B. subtilis* mutants at different potassium concentrations, the bacteria were inoculated in LB medium and precultured in MSSM medium with 0.1 mM KCl. The cultures were grown until exponential phase, harvested, washed three times in MSSM basal salts solution before the $OD_{600}$ was adjusted to 1.0. Dilution series were then pipetted onto MSSM plates with ammonium or glutamate and 0.1 or 5 mM potassium.

Complementation in *E. coli* was assayed by transformation of strain JM109 that is unable to grow with glutamate as the single source of carbon and nitrogen. For this purpose, the strain was transformed with the complementation plasmid pGP2987 (*aimA*) and the empty vector control pWH844. Growth of the transformants was assessed on M9 minimal medium plates containing either ammonium and glucose or glutamate as carbon and nitrogen source [73].

## DNA manipulation and genome sequencing

Transformation of *E. coli* and plasmid DNA extraction were performed using standard procedures [88]. All commercially available plasmids, restriction enzymes, T4 DNA ligase and DNA polymerases were used as recommended by the manufacturers. Chromosomal DNA of *B*.

*subtilis* was isolated as described [89]. *B. subtilis* was transformed with plasmid and genomic DNA according to the two-step protocol [89].

To identify the mutations in the suppressor mutants, their genomic DNA was subjected to whole-genome sequencing [92]. Briefly, the reads were mapped on the reference genome of *B. subtilis* 168 (GenBank accession number: NC_000964) [93] using the Geneious software package (Biomatters Ltd., New Zealand) [94]. Single nucleotide polymorphisms were considered as significant when the total coverage depth exceeded 25 reads with a variant frequency of ≥90%. All identified mutations were verified by PCR amplification and Sanger sequencing.

## Construction of mutant strains by allelic replacement

Gene deletion was achieved by transformation of *B. subtilis* 168 with a PCR product constructed using oligonucleotides to amplify DNA fragments flanking the target genes and an appropriate intervening resistance cassette as described previously [95]. The mutation *yfkC*-R42W was introduced by amplification of regions containing and flanking the mutation from strain GP2837, respectively, and a kanamycin resistance cassette was introduced downstream of *yfkC* in the non-coding region between *yfkC* and *yfkD* (terminator region of both genes present). The integrity of the regions flanking the integrated resistance cassette was verified by sequencing PCR products of about 1,000 bp amplified from chromosomal DNA of the resulting mutant strains.

## Plasmids

Construction of a reporter fusion of the *kimA* promoter region with the promoterless *lacZ* gene was done using the plasmid pAC6 [96]. The promoter fragment was cloned between the EcoRI and BamHI sites of pAC6. The resulting plasmid was pGP2912. The plasmid was linearized with ScaI and transformed into *B. subtilis* 168 for chromosomal integration into the *amyE* locus. For *E. coli* complementation assays, the *aimA* gene was amplified using the primer pair LK167 (5' AAAGGATCCATGAATCAATTGCATCGAAGAATGGGAACGTTTT)/ AK324 (5' TTTGTCGACTTATTCTTTTCCGGCAGCAGCTTCTG) and cloned between the BamHI and SalI sites of the expression vector pWH844 [74]. The resulting plasmid was pGP2987.

## Determination of apparent glutamate transporter kinetics

The growth characteristics of *B. subtilis* glutamate transporter mutants were used to determine the single glutamate transporter kinetics. The mutants were either positive for *gltT* (Δ*gltP*::*cat* Δ*aimA*::*kan*) or *aimA* (Δ*gltT*::*kan* Δ*gltP*::*cat*) or lacked both (Δ*gltT*::*kan* Δ*aimA*::*phleo* Δ*gltP*:: *cat*). The nitrogen source was used as the limiting factor, so that transport of glutamate was the essential factor for growth. The bacteria were inoculated in LB medium and precultured in MSSM medium (ammonium) with 0.1/5 mM KCl. The cultures were grown until exponential phase and the cells were washed three times in MSSM basal salts solution. Afterwards, the cells were adjusted to $OD_{600}$ 1.0 and used to inoculate a 96 well plate (Microtest Plate 96 Well, F, Sarstedt) containing the MSSM medium with the respective glutamate and potassium concentrations. The growth was tracked in an Epoch 2 Microplate Spectrophotometer (BioTek Instruments) at 37°C with linear shaking at 237 cpm (4 mm) for 20 h, and optical density at 600 nm was measured in 10 min intervals. The exponential growth phase was used to determine the growth rate μ [$h^{-1}$]. μ was calculated from $t_1$ (early exponential phase) and $t_2$ (late exponential phase) and the corresponding $OD_1$ and $OD_2$ values. The resulting growth rates were then plotted against the glutamate concentrations. This allowed fitting to the Monod equation and calculation of $V_{max}$ ($h^{-1}$) and the apparent $K_S$ (mM glutamate) using the solver tool of Excel 2012 (Microsoft). Experiments were repeated with three biological replicates.

## Metabolite analysis

Bacterial cells were cultivated in MSSM minimal medium with ammonium and 0.1 mM KCl until late exponential growth phase. 4 ml of each culture were harvested by filtration as described previously [97]. The filter was washed three times with 0.9% NaCl and transferred to 4 ml extraction solution (60% ethanol containing the internal $^{15}$N L-glutamic acid standard) and stored over night at -20˚C. The extraction of metabolites and the derivatization was performed as described previously [92]. For the analysis of the medium, the cells were cultivated in the same medium until late exponential growth phase and harvested by centrifugation (8,500 rpm, 10 min). 100 μl of the supernatant were used for further analysis. $^{15}$N L-glutamic acid standard was added and the sample evaporated by nitrogen stream. The sample was dissolved in 15 μl Acetonitrile and 15 μl of N-methyl-N-(trimethylsilyl)trifluoroacetamide was added for derivatization. Analysis by GC-MS was performed with previously used equipment and settings [98].

## Transcriptome analysis

Wild type and the c-di-AMP-deficient strains GP2222 Δ*dac* and GP2223 Δ*dac nhaK*-S187F were grown in MSSM medium with ammonium or glutamate and low (0.1 mM) or high (5 mM) potassium concentrations. The strains were harvested by centrifugation (10.397 × g, 1 min, 4˚C) at mid exponential phase (OD$_{600}$ of 0.4–0.6). A total of two independent biological replicates were included. The pellets were frozen immediately in liquid nitrogen and stored at −80˚C. Isolation of the total RNA was done as described previously [99,100], and the quality of the RNA preparations was assessed by means of an Agilent 2100 Bioanalyzer according to the manufacturer's instructions. 5 μg of total RNA were used for cDNA synthesis. The synthesis and fluorescence labeling of cDNA followed a strand-specific method using the FairPlay III Microarray Labeling Kit (Agilent Technologies, Santa Clara, CA, USA) and actinomycin D (Calbiochem) [101]. 200 ng of Cy3-labeled cDNA were hybridized to the microarray following Agilent's hybridization, washing and scanning protocol (One-Color Microarray-based Gene Expression Analysis, version 5.5). Data were extracted and processed using the Feature Extraction software (version 11.5.1.1). An aggregated expression value was computed for each annotated coding sequence and previously identified RNA feature [100]. Gene-level intensities were scaled based on the intensity values of ten different *in vitro*-synthesized transcripts contained in the One Color RNA Spike-In kit (Agilent Technologies) in order to account for technical variation associated with sample processing. Genes with at least 2.5-fold difference in expression levels of the wild type at the different conditions or between wild type and GP2222 mutant were considered significantly affected. The graphical presentation of the results was generated using GraphPad Prism version 8.0.0 for Windows, GraphPad Software, San Diego, California USA, www.graphpad.com (see S11 Fig).

To analyze the regulatory network, we compared the expression levels of the wild type strain and the GP2222 mutant at ammonium 0.1 mM KCl. The genes were clustered in operons according to their genetic localization, as described in *Subti*Wiki [51]. Additionally, gathered operons were grouped by their corresponding documented regulator/s and metabolism. The visual representation of the regulatory network was generated with Cytoscape and GIMP softwares [102,103].

## Quantitative Real-Time-PCR

For RNA isolation, the cells were grown in MSSM minimal medium containing ammonium or glutamate, and 0.1 mM or 5 mM KCl to an OD$_{600}$ of 0.4 to 0.6 and harvested. Preparation of total RNA was carried out as described previously [104]. cDNAs were synthesized using a

One-Step reverse transcription-PCR (RT-PCR) kit (Bio-Rad) [95]. Quantitative RT-PCR (qRT-PCR) was carried out on an iCycler instrument (Bio-Rad) following the manufacturer's instructions. The primers used for amplification are listed in S3 Table. The *alaS* and *gyrA* genes were used as internal controls, as their expression has been shown to be stable [105]. Data analysis and the calculation of expression ratios as fold changes were performed as described previously [95]. qRT-PCR experiments were performed in triplicates.

## Supporting information

**S1 Text. Global gene expression in the *B. subtilis* wild type strain in response to the potassium concentration and the nitrogen source.**
(DOCX)

**S1 Table. Transcriptomic data for the genes that are most strongly affected by the absence of c-di-AMP (Ammonium 0.1 mM KCl).**
(DOCX)

**S2 Table. Transcriptomic data of genes involved in c-di-AMP, potassium, and glutamate homeostasis.**
(DOCX)

**S3 Table. Oligonucleotides used for qRT-PCR analyses.**
(DOCX)

**S4 Table. Numerical data that underlies graphs or summary statistics.**
(XLSX)

**S1 Fig. Determination of minimal inhibitory concentration of glutamate for Δdac mutant.** Growth of *B. subtilis* 168 and GP2222 (Δ*dac*) was assessed on MSSM medium with ammonium and 0.1 mM and the indicated amount of glutamate (%).
(TIF)

**S2 Fig. Acquisition of beneficial mutations allows growth of a c-di-AMP deficient strain in the presence of glutamate.** Growth assay of *B. subtilis* wild type, GP2222 (Δ*dac*), and the isolated glutamate suppressor mutants. *B. subtilis* strains were cultivated in MSSM minimal medium with 0.1 mM KCl and ammonium. The cells were harvested, washed, and the OD$_{600}$ was adjusted to 1.0. Serial dilutions were dropped onto MSSM minimal plates with the indicated potassium concentration and ammonium or glutamate, or on LB plates.
(TIF)

**S3 Fig. Two-step adaptation to complex medium.** Growth assay of *B. subtilis* wild type, GP2779 (Δ*dac* Δ*darB*), GP2420 (Δ*dac* Δ*darA*) and the isolated glutamate suppressor mutants. *B. subtilis* strains were precultivated in LB medium. The cells were harvested, washed, and the OD$_{600}$ was adjusted to 1.0. Serial dilutions were dropped onto MSSM minimal plates with 0.1 mM KCl and ammonium or on LB plates.
(TIF)

**S4 Fig. Validation of the c-di-AMP dependent regulation of genes involved in respiration and fermentation (A) and potassium uptake (B).** *B. subtilis* wild type and Δ*dac* were cultivated in MSSM medium with ammonium and 0.1 mM KCl (A) or in MSSM medium with ammonium or glutamate and 0.1 mM or 5 mM KCl (B) and RNA was extracted and quantified by qRT-PCR. Changes in expression are represented as fold changes compared to the wild type grown with Ammonium and 0.1 mM KCl. The tables show a comparison of the ratios as

determined by the transcriptomic analysis and qRT-PCR.
(TIF)

**S5 Fig. Phospholipid biosynthesis in B. subtilis.** In the suppressor screen with glutamate (Table 1) and LB (Fig 2), mutations in *accA*, *accC*, and *plsC*, which are involved in the synthesis of phospholipids, were obtained. All mutations were single amino acid substitutions from independently isolated clones. All mutations are located in highly conserved regions of AccA and AccC (Acetyl-CoA carboxylase, S7 Fig) and PlsC (acyl-ACP:1-acylglycerolphosphate acyl-transferase, S8 Fig) close to the active center.
(TIF)

**S6 Fig. Mutations in the acetyl-CoA carboxylase subunits.** Mutations obtained in the suppressor screen with glutamate (Table 1) and LB (Fig 2). All mutations were single amino acid substitutions from independently isolated clones. Both mutations are located in highly conserved regions of AccA and AccC (hatched boxes) close to the active center.
(TIF)

**S7 Fig. Mutations in the acyl-ACP:1-acylglycerolphosphate acyltransferase.** Mutations in the *plsC* gene were obtained in the suppressor screen with glutamate (Table 1). All mutations were single amino acid substitutions from independently isolated clones. All mutations are located in functionally important regions of the enzyme.
(TIF)

**S8 Fig. Mutations in the mechanosensitive channel YfkC.** Mutations in the *yfkC* gene were obtained in the suppressor screen with glutamate (Table 1). The amino acid substitution Arg42 to Trp was observed in three independently isolated clones. In *E. coli* localization of S58 was observed to be highly dependent on the closing state of the channel, thus, it appears likely that amino acid residue 42 in *B. subtilis* is located close to the gate.
(TIF)

**S9 Fig. Mutations in the elongation factor G (FusA).** Mutations in the *fusA* gene were obtained in the suppressor screen of GP3054 (Δ*dac* Δ*aimA*) with glutamate (Fig 1). The amino acid substitution Ala21 to Thr was observed in one suppressor. Ala21 is adjacent to Asp20, an amino acid residue that binds the Mg ion and is therefore crucial for GTPase activity of the protein. In the two other suppressors Ala579 was mutated to Val. This amino acid residue is located in the conserved loop II that is required for proper translocation.
(TIF)

**S10 Fig. Effect of the deletion of the glutamate transporters in a c-di-AMP deficient strain.** Growth assay of *B. subtilis* wild type, GP2222 (Δ*dac*), GP3054 (Δ*dac* Δ*aimA*), GP2248 (Δ*dac* Δ*gltT*) and the isolated glutamate suppressor mutants. *B. subtilis* strains were cultivated in MSSM minimal medium with 0.1 mM KCl and ammonium. The cells were harvested, washed, and the $OD_{600}$ was adjusted to 1.0. Serial dilutions were dropped onto MSSM minimal plates with the indicated potassium concentration and ammonium or glutamate, or on LB plates.
(TIF)

**S11 Fig. Heatmap representation of the transcriptomic data.** Representation of expression patterns of the most differentially expressed genes without c-di-AMP (see S1 Table). Blue and red indicate up- and downregulated expression, respectively. Color density represents the level of fold change. The figure was prepared with GraphPad Prism 8.
(TIF)

**S12 Fig. Potassium and glutamate influenced promoters.** The expression of the high-affinity potassium transporter *kimA* (A) and the glutamate synthase *gltA* (B) was assessed by fusion of the promoter to the reporter gene *lacZ*. The cells harboring the promoter-fusion were cultivated with ammonium or glutamate at 0.1 mM KCl (*kimA*) or ammonium and different potassium concentrations (0.1, 5, 20 mM; *gltA*). Promoter activity was analyzed by quantification of β-galactosidase activity. Statistical analysis was performed using a one-way ANOVA, followed by Tukey's multiple comparisons test (**** $P < 0.0001$).
(TIF)

## Acknowledgments

We are grateful to Anika Klewing, Marina Rodnina, Fabian Commichau, and Frank Peske for helpful discussion. We like to thank Felix Mehne and Martin Weiss for the help with strain constructions and the initial characterization of a suppressor mutant, respectively.

## Author Contributions

**Conceptualization:** Larissa Krüger, Jan Gundlach, Jörg Stülke.

**Data curation:** Hermann Rath, Tiago Pedreira, Ulrike Mäder.

**Funding acquisition:** Jörg Stülke.

**Investigation:** Larissa Krüger, Christina Herzberg, Hermann Rath, Till Ischebeck, Anja Poehlein, Jan Gundlach, Ulrike Mäder.

**Methodology:** Larissa Krüger, Christina Herzberg.

**Project administration:** Jörg Stülke.

**Supervision:** Rolf Daniel, Uwe Völker, Jörg Stülke.

**Writing – original draft:** Larissa Krüger, Jörg Stülke.

**Writing – review & editing:** Larissa Krüger, Jan Gundlach, Ulrike Mäder, Jörg Stülke.

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
