## [Decision Letter · Decision Letter 0]

28 Sep 2020

Dear Dr Stülke,

Thank you very much for submitting your Research Article entitled 'Essentiality of c-di-AMP in Bacillus subtilis: Bypassing mutations converge in potassium and glutamate homeostasis' to PLOS Genetics. Your manuscript was fully evaluated at the editorial level and by independent peer reviewers. The reviewers appreciated the attention to an important problem, but raised some substantial concerns about the current manuscript. Based on the reviews, we will not be able to accept this version of the manuscript, but we would be willing to review again a much-revised version. We cannot, of course, promise publication at that time.

If you decide to revise the manuscript for further consideration at PLOS Genetics, please aim to resubmit within the next 60 days, unless it will take extra time to address the concerns of the reviewers, in which case we would appreciate an expected resubmission date by email to plosgenetics@plos.org.

[LINK]

We are sorry that we cannot be more positive about your manuscript at this stage. Please do not hesitate to contact us if you have any concerns or questions.

Yours sincerely,

Jue D. Wang

Associate Editor

PLOS Genetics

Josep Casadesús

Section Editor: Prokaryotic Genetics

PLOS Genetics

Reviewer's Responses to Questions

**Comments to the Authors:**

Reviewer #1: c-di-AMP is an essential signaling molecule under standard lab growth conditions, although certain species lacking c-di-AMP can grow when plated on minimal media or when grown on complex media anaerobically. Previous work on this nucleotide has enabled the isolation of suppressor mutants that improve growth, with mutations affecting osmolyte or glutamate uptake. Here the authors do in-depth suppressor screening to uncover more about the importance of c-di-AMP for balancing potassium and glutamate homeostasis. Suppressor mutants are isolated on minimal media with toxic levels of glutamate or on complex media, both anaerobically and then aerobically. The authors conclude that the adaptation to these growth conditions requires the acquisition of multiple mutations, which frequently affect potassium, osmolyte or glutamate metabolism, but also find a connection to phospholipid biosynthesis. This manuscript showcases a lot of work that will certainly add to the field, and as such deserves publication. I have two recommendations that I feel would strengthen the manuscript and then some minor comments.

Major

• On line 184 the authors state “it seems likely that more than one mutation is required to achieve suppression”. In the methods it explains that the whole genome sequencing was performed on suppressors that had been re-isolated 3 times after the original colony appeared. Therefore it is currently not clear whether all of the mutations in Table 1 arose in the original suppressor colony or whether they were introduced during re-isolation to improve growth. This is something I remember reading about in Corrigan et al 2011 Plos Pathogens paper on c-di-AMP. An easy way to confirm the need for multiple mutations for suppression is to perform PCR on chromosome from the original colony to confirm that all mutations are present. It may also highlight a smaller number of mutations that are the most critical for suppression of the lack of c-di-AMP.

• Fig 4 shows a strong growth phenotype with suppressor strain GP2837, due to mutations in PlsC and YfkC. However the single YfkC* mutation does not improve growth, whereas the combination of YfkC* and PlsC L48P restores to near WT. Has the PlsC L48P mutation been tested alone? It may be that PlsC L48P is responsible for the restoration in growth and plays a larger role than YfkC*.

Minor

• The introduction, although quite long, should introduce the B. subtilis target proteins in greater detail. For example, KtrC is mentioned frequently in the results and should be introduced as the homologue of the KtrA system. DarA and DarB are introduced as “signal transduction proteins” but this is not very informative. One of two sentences on the targets would aid the reader.

• Line 641: what was the rationale behind incubating at 42oC to isolate suppressors? Wouldn’t this just stress the bacteria more?

• Please use a footnote for Table 1 to define the scoring system.

• Using a heat map for Table S1 would really help in visualizing what has gone up or down.

• Line 178 – NhaK – what is this protein and how does it affect K+ homeostasis?

• Line 376 and 377 – this statement refers to Fig S4, not S5. Similarly line 387 should refer to Fig S5, not S6.

• Are there error bars and stats for Fig 5e? Please state in all of the legends how many repeats were performed and include statistics were appropriate.

Reviewer #2: The manuscript by Kruger et al. examined the Bacillus subtilis dac mutant, deleted for all three diadenylate cyclases and therefore devoid of c-di-AMP, to elucidate the mechanisms of glutamate toxicity to this mutant and the mechanisms of c-di-AMP essentiality in Bacillus growth in rich media (LB). Through multiple genetic screens, the authors isolated many suppressor mutants for each phenotype, and presented thoughtful analyses of each mutation. The identification of these mutations is useful to understand c-di-AMP signaling. The mechanisms for glutamate toxicity appear largely related to potassium homeostasis. The mechanisms for c-di-AMP essentiality in LB are unclear. There are some instances where a few more control experiments would further support the conclusions, and the clarity of the manuscript could be improved with some re-organization and provision of model figures.

My scientific comments are as follows:

1. Although the suppressor screens revealed very interesting mutations, many remain to be validated, with a notable example of plsC mutations. If reduced phospholipid synthesis suppresses glutamate toxicity as the authors interpreted, can this phenotype be reversed upon expression of WT plsC in the relevant suppressor mutants? Similar complementation should also be feasible for other loss-of-function mutations, such as ndhF, fliP, etc.

2. Related to my comment #1 above, I suggest that the transcriptomic data be validated by qPCR or transcriptional fusion reporter assays, at least for a select set of genes that the authors discussed in great details or were presented in the main figures, ie. gltA/aimA/gltP expression (Figure 5E), ndhF, ldh.

3. What is the approximate glutamate minimal inhibitory concentration in the dac mutant? The suppressor screen for glutamate toxicity was performed with 1% glutamate (equivalent to 70mM?), which seemed rather high considering that later experiments for glutamate growth (shown in Figure 5) were done for 20mM glutamate or less? On a related note, perhaps glutamate concentrations used would affect which glutamate transporters would be inactivated to suppress glutamate toxicity?

4. The screen for dac suppressors in LB revealed several interesting phenotypes, but also somewhat complicated the interpretation of glutamate toxicity suppressors isolated from MSSM.

- Some glutamate resistant mutants in Table 1 grew in LB. Was growth observed under aerobic or anaerobic condition? If these mutants can grow aerobically in LB, how do the authors reconcile this observation with the LB suppressor screen results and discussion related to Figure 2?

- Was it surprising that glutamate transporters were not identified in the LB suppressor screen? What’s the glutamate content in LB?

5. The authors alluded to the essentiality of c-di-AMP in aerobic respiration, through stepwise suppressor screens and transcriptomic analysis, and as also discussed in lines 584-598. I think some more controls should be done to establish this possible regulation and distinguish it from glutamate toxicity or other nutrient effects.

- Related to results in Table 1 and discussion point in lines 585-586: Is glutamate and potassium toxicity dependent on oxygen? Does complementation with WT ndhF restore glutamate toxicity or alter aerobic growth?

- Does the Bacillus dac mutant grow differently under aerobic and anaerobic conditions in MSSM?

- Can you expand your discussion on why LB might be more permissive to anaerobic growth than aerobic growth, at least for the dac darA/B mutants?

- Based on the decreased expression of both ldh-lctP and cydABCD, what conclusions can be drawn about aerobic respiration vs fermentation activities in dac?

6. The observation that mutations in aimA were frequently isolated in the dac suppressor screen, as well as data from Figure S8 (dac aimA grew but dac gltT did not), might also indicate that the dac mutant is sensitive to serine. Have the authors examined this possibility and discerned it from glutamate toxicity?

Minor comments:

7. The authors should consider showing schematic diagrams of relevant pathways to better predict the mechanisms of suppressor phenotypes. For instance, an abbreviated schematic diagram of phospholipid biosynthesis showing PlsC and AccA-C would greatly clarify data interpretation in lines 367-390. An overall model figure summarizing the mechanisms of glutamate toxicity and c-di-AMP essentiality in LB would also be a fantastic addition.

8. Was gene expression in the transcriptomic data set normalized to a housekeeping gene?

9. Section on LB suppressor screen (lines 214-277) is perhaps best moved to the end of the Results since all other sections address glutamate toxicity.

Reviewer #3: Review of “Essentiality of c-di-AMP in Bacillus subtilis: Bypassing mutations converge in potassium and glutamate homeostasis" (PGENETICS-D-20-01328)

This manuscript describes the isolation and analysis of suppressor mutants of a Bacillus subtilis mutant devoid of all three diadenylate cyclase genes. Whilst suppressors were unable to be obtained on rich media (unless darA or darB were also inactivated and incubation was under anaerobic conditions), several suppressors could be isolated on minimal media containing glutamate. Genome analysis revealed multiple mutations in most suppressors, with most occurring in genes encoding a novel glutamate transporter (YbeC/AimA), several different potassium transporters, phospholipid biosynthesis genes and a mechanosensitive exporter (YfkC). Further characterization revealed that AimA is a low affinity glutamate transporter while GltT is a high affinity glutamate transporter. Work also showed that potassium levels in the media affect glutamate transport by GltT (but not AimA), which would allow balanced regulation of these major ions in the cell. In addition, expression of AimA is increased in the absence of c-di-AMP, which provides an explanation of why the diadenylate cyclase mutant is sensitive to glutamate (however the mechanism of this regulation was not explored). Overall the work is interesting and provides several new insights into why c-di-AMP is essential for bacteria. The paper is clearly written and the complex findings in the suppressors are well presented.

Limitations of the work include where it describes several suppressors with mutations in genes affecting phospholipid biosynthesis and a mechanosensitive channel, however their roles in restoring growth in mutants unable to synthesize c-di-AMP is not clear. There is quite a lot of speculation that the YfkC mutation is a gain-of-function change and may lead to enhanced glutamate export, however this was not explored more. This would make sense and would be a novel finding and in alignment with the theme of the paper, which is focussed primarily on glutamate homeostasis. To strengthen the paper it would be worthwhile to determine if intracellular glutamate levels are reduced in cells carrying the variant YfkC protein (this could perhaps be performed in the wild-type strain carrying the yfkC variant).

Specific comments:

- Line 120 – should glutamate be glutamine?

- Table 1 – should define the criteria for the +++, ++, + and – scores (is it simply colony size?)

- Line 189 and after – change YbeC to AimA (since it was renamed in line 140)

- Lines 402-404 – the reference to Figure 4 here doesn’t strongly support the statement (the rapid appearance of suppressors in the nhaK background strain). I can see one colony on the 10E-1 dilution, but none on the 10E0 dilution, suggesting that suppressors are still relatively uncommon. I suggest to remove the reference to this figure in this sentence (or provide some stronger support from an additional experiment)

- Line 575 – speculation of the gain-of-function change in YfkC is not strongly supported by data.

- Figure 1 – there appear to be some listed mutations missing for the suppressors (GP3102 and GP3103) of the delta-dac/delta-aimA mutant.

**Have all data underlying the figures and results presented in the manuscript been provided?**

Reviewer #1: Yes

Reviewer #2: Yes

Reviewer #3: Yes

PLOS authors have the option to publish the peer review history of their article (what does this mean?). If published, this will include your full peer review and any attached files.

Reviewer #1: No

Reviewer #2: No

Reviewer #3: **Yes: **Mark S Turner

---

## [Decision Letter · Decision Letter 1]

14 Dec 2020

Dear Dr Stülke,

We are pleased to inform you that your manuscript entitled "Essentiality of c-di-AMP in Bacillus subtilis: Bypassing mutations converge in potassium and glutamate homeostasis" has been editorially accepted for publication in PLOS Genetics. Congratulations!

Yours sincerely,

Jue D. Wang

Associate Editor

PLOS Genetics

Josep Casadesús

Section Editor: Prokaryotic Genetics

PLOS Genetics

Comments from the reviewers (if applicable):

Reviewer's Responses to Questions

**Comments to the Authors:**

Reviewer #1: All of my comments have been addressed and the additions to figure 4 make this a stronger paper. Overall it is well written and adds a lot to our knowledge of c-di-AMP signalling.

One comment to address - I can't find reference to Fig S11 or S12 in the text

Reviewer #2: Since the last submission, the authors have addressed all reviewers' comments and critiques, and have added valuable data. Overall, this is a much improved manuscript and provides a much clearer picture of the molecular bases for c-di-AMP essentiality in Bacillus subtilis. Glutamate toxicity in the dac mutant is strongly linked to potassium transport. The inability of the dac mutant to grow in rich medium (LB) is complex but is thoroughly discussed. This manuscript will be a valuable reference for other c-di-AMP-producing bacteria.

Reviewer #3: The authors have carried out a significant amount of additional experimental work and modifications to the text, which have improved the manuscript. In particular, the new data confirming the role of YfkC in glutamate export is a very nice addition.

**Have all data underlying the figures and results presented in the manuscript been provided?**

Reviewer #1: Yes

Reviewer #2: Yes

Reviewer #3: Yes

PLOS authors have the option to publish the peer review history of their article (what does this mean?). If published, this will include your full peer review and any attached files.

Reviewer #1: No

Reviewer #2: No

Reviewer #3: No

**Data Deposition**

http://datadryad.org/submit?journalID=pgenetics&manu=PGENETICS-D-20-01328R1

**Press Queries**

---

## [Editor Report · Acceptance letter]

18 Jan 2021

PGENETICS-D-20-01328R1 

Essentiality of c-di-AMP in Bacillus subtilis: Bypassing mutations converge in potassium and glutamate homeostasis 

Dear Dr Stülke, 

We are pleased to inform you that your manuscript entitled "Essentiality of c-di-AMP in Bacillus subtilis: Bypassing mutations converge in potassium and glutamate homeostasis" has been formally accepted for publication in PLOS Genetics! Your manuscript is now with our production department and you will be notified of the publication date in due course.

With kind regards,

Melanie Wincott

PLOS Genetics

On behalf of:
